# PDX1^LOW MAFA^LOW β-cells contribute to islet function and insulin release

Daniela Nasteska[1,2,3], Nicholas H. F. Fine [1,2,3], Fiona B. Ashford[1,2,3], Federica Cuozzo [1,2,3], Katrina Viloria[1,2,3], Gabrielle Smith[1,3], Aisha Dahir[1,3], Peter W. J. Dawson[4,5], Yu-Chiang Lai[1,4,5], Aimée Bastidas-Ponce[6,7,8,9], Mostafa Bakhti [6,7,8], Guy A. Rutter [10,11], Remi Fiancette[12], Rita Nano[13,14], Lorenzo Piemonti [13,14], Heiko Lickert [6,7,8,9], Qiao Zhou[15], Ildem Akerman [1,3] & David J. Hodson [1,2,3✉]

Transcriptionally mature and immature β-cells co-exist within the adult islet. How such diversity contributes to insulin release remains poorly understood. Here we show that subtle differences in β-cell maturity, defined using PDX1 and MAFA expression, contribute to islet operation. Functional mapping of rodent and human islets containing proportionally more PDX1^HIGH and MAFA^HIGH β-cells reveals defects in metabolism, ionic fluxes and insulin secretion. At the transcriptomic level, the presence of increased numbers of PDX1^HIGH and MAFA^HIGH β-cells leads to dysregulation of gene pathways involved in metabolic processes. Using a chemogenetic disruption strategy, differences in PDX1 and MAFA expression are shown to depend on islet $Ca^{2+}$ signaling patterns. During metabolic stress, islet function can be restored by redressing the balance between PDX1 and MAFA levels across the β-cell population. Thus, preserving heterogeneity in PDX1 and MAFA expression, and more widely in β-cell maturity, might be important for the maintenance of islet function.

[1] Institute of Metabolism and Systems Research (IMSR), University of Birmingham, Birmingham, UK. [2] Centre of Membrane Proteins and Receptors (COMPARE), University of Birmingham and University of Nottingham, Midlands, UK. [3] Centre for Endocrinology, Diabetes and Metabolism, Birmingham Health Partners, Birmingham, UK. [4] School of Sport, Exercise and Rehabilitation Science, University of Birmingham, Edgbaston, UK. [5] MRC-Versus Arthritis Centre for Musculoskeletal Ageing Research, University of Birmingham, Edgbaston, UK. [6] Institute of Diabetes and Regeneration Research, Helmholtz Zentrum München, D-85764 Neuherberg, Germany. [7] German Center for Diabetes Research (DZD), D-85764 Neuherberg, Germany. [8] Institute of Stem Cell Research, Helmholtz Zentrum München, D-85764 Neuherberg, Germany. [9] Technical University of Munich, School of Medicine, Munich, Germany. [10] Section of Cell Biology and Functional Genomics, Division of Diabetes, Endocrinology, and Metabolism, Department of Metabolism, Reproduction, and Digestion, Imperial College London, London, UK. [11] Lee Kong Chian School of Medicine, Nanyang Technological University, Nanyang, Singapore. [12] Institute of Immunology & Immunotherapy, College of Medical and Dental Sciences, University of Birmingham, Birmingham, UK. [13] San Raffaele Diabetes Research Institute, IRCCS Ospedale, San Raffaele, Italy. [14] Vita-Salute San Raffaele University, Milan, Italy. [15] Division of Regenerative Medicine, Department of Medicine, Weill Cornell Medical College, New York, NY, USA. ✉email: d.hodson@bham.ac.uk

Type 2 diabetes mellitus (T2DM) occurs when β-cells are unable to release enough insulin to compensate for insulin resistance. At the cellular level, glucose-regulated insulin secretion depends upon generation of ATP/ADP, closure of ATP-sensitive $K^+$ ($K_{ATP}$) channels, opening of voltage-dependent $Ca^{2+}$ channels (VDCC) and exocytosis of insulin granules[1]. At the multicellular level, insulin release is a tightly controlled process, requiring hundreds of β-cells throughout the islet to coordinate their activities in response to diverse stimuli including glucose, incretins and fatty acids[2,3].

Our current understanding of the mechanisms underlying insulin release is mainly derived from experiments in single β-cells or measures averaged across the entire β-cell complement. However, such studies, which generally view β-cells as a tightly coupled system, are difficult to reconcile with the known heterogeneous nature of β-cell identity and function. Based on transcriptomic[4,5] and protein signatures[6], marker analyses[7–9], glucose-responsiveness[10,11], reporter imaging[12–15] or single molecule hybridization[16], β-cell subpopulations have been shown to exist with altered maturity states, metabolism, electrical activity, insulin secretion and proliferative capacity (reviewed in[17,18]). Of note, β-cell subpopulations are highly plastic. During aging and T2DM, β-cells with reduced maturity, metabolism and insulin secretion, but enhanced proliferative capacity, typically increase in proportion in both rodent and human[4,7,8]. At the same time, there is an increase in the number of mature, secretory β-cells that display poorer proliferative capacity[6,7]. Thus, the adult islet houses highly plastic mature and immature β-cell subpopulations whose co-existence might be important for balancing renewal with the need for insulin release.

Mature β-cells are generally thought to contribute the most to islet function, since they comprise ~70–90% of the β-cell population, express higher levels of insulin, glucose transporter, glucokinase and maturity genes, and mount normal ATP/ADP and $Ca^{2+}$ responses to stimulus (reviewed in[19]). By contrast, immature β-cells are in the minority, show poor glucose-responsiveness and are less secretory[4,7,8,14,19]. However, β-cell subpopulations that disproportionately influence islet responses to glucose have recently been identified in situ and in vivo[20–22]. One of the subpopulations, termed hubs, was found to display lowered expression of β-cell maturity markers and insulin, but increased expression of glucose-sensing enzymes, including glucokinase[21,22]. These studies provide the first glimpse that immature cells with similar characteristics might contribute to the regulation of insulin release across the islet.

We hypothesized that transcriptionally immature β-cells (PDX1$^{LOW}$/MAFA$^{LOW}$) belong to a highly functional subpopulation, able to overcome their relative deficiencies by interacting with their more mature counterparts to drive insulin release. In this work, we use recombinant genetic and chemogenetic disruption strategies to alter the balance of PDX1$^{LOW}$/MAFA$^{LOW}$:PDX1$^{HIGH}$/MAFA$^{HIGH}$ β-cells in the islet. An increase in the proportion of PDX1$^{HIGH}$/MAFA$^{HIGH}$ β-cells leads to defective ionic and metabolic fluxes, dysregulation of genes involved in metabolism, as well as impaired insulin secretion. Heterogeneity in PDX1 and MAFA is encoded at the individual β-cell level by the islet $Ca^{2+}$ signaling network, and maintaining a balance between PDX1$^{LOW}$/MAFA$^{LOW}$:PDX1$^{HIGH}$/MAFA$^{HIGH}$ β-cells restores $Ca^{2+}$ fluxes during metabolic stress. Together, these results show that differences in PDX1 and MAFA levels, and more broadly in β-cell maturity, contribute to islet function.

## Results

### Generation of islets with proportionally more PDX$^{HIGH}$/MAFA$^{HIGH}$ β-cells. We first generated and validated an

overexpression model to alter the balance between immature and mature β-cells throughout the population. Here, immature β-cells are operationally defined as expressing low levels of the transcription factors PDX1 and MAFA based upon fluorescent immunostaining. Islets were transduced with control adenovirus containing PATagRFP (β normal; B-NORM) or a polycistronic construct encoding NEUROG3/PDX1/MAFA (Ad-M3C) (β mature; B-MAT). The M3C construct is well-validated[23,24], a TetO mouse possessing the same construct exists[25], and driving multiple transcription factors using the same promoter avoids heterogeneous expression profiles. Ad-M3C was able to drive exogenous *Neurog3*, *Pdx1* and *Mafa* expression (Fig. 1a), expected to occur predominantly in the first two layers of the islet where functional imaging takes place. Native gene expression levels remained unchanged for *Neurog3* and *Mafa*, but ~ 25% lower for *Pdx1*, consistent with the absence of positive autoregulation seen with Pdx1-fluorophore constructs[26].

Analyses of individual β-cells in intact islets showed a non-Gaussian distribution of PDX1 and MAFA protein fluorescence intensities in B-NORM islets, which we arbitrarily define as PDX1$^{LOW}$/MAFA$^{LOW}$ and PDX1$^{HIGH}$/MAFA$^{HIGH}$ using a 15% cut-off (i.e. the bins spanning 0–15 normalized PDX1/MAFA intensity units) (Fig. 1b–d). By contrast, there was a significant reduction in the proportion of cells occupying the lowest 15 % of bins for detectable PDX1 and MAFA expression in B-MAT islets (Fig. 1b–d) (Supplementary Fig. 1 and Supplementary Fig. 2a–c).

Quantification was repeated using DAPI staining for normalization (Supplementary Fig. 2d and e), or taking into account only INS+ cells (Supplementary Fig. 2f), with similar results. Analysis of PDX1, MAFA and INS immunoreactivity in B-NORM islets showed a positive association across hundreds of cells examined, suggesting that PDX1$^{LOW}$ and MAFA$^{LOW}$ cells are functionally immature (Fig. 1e–g). While very low levels of NEUROG3 could be detected in B-MAT islets (Supplementary Fig. 2g), a progenitor signature was not detected at the transcriptomic level (see below). A generalized PDX1 overexpression across the β-cell population was unlikely given that the mean fluorescence intensity was only slightly (~20%) increased in B-MAT islets (Supplementary Fig. 2h–j), consistent with the reported 2-fold increase in PDX1 expression in PDX$^{LOW}$ cells (Fig. 1b–d) (Supplementary Fig. 1 and Supplementary Fig. 2a–c). Preferential overexpression in PDX1$^{LOW}$/MAFA$^{LOW}$ (immature) β-cells was confirmed using Pdx1-BFP reporter islets[26], which read out endogenous *Pdx1* levels. Quantification of PDX1 and BFP levels in the same cells revealed a strong positive linear correlation in B-NORM islets. However, the correlation was weaker (and slope less steep) in B-MAT islets due to transition of a subpopulation of BFP$^{LOW}$ cells to a PDX1$^{HIGH}$ state (Fig. 1h). Supporting this finding, BFP$^{LOW}$ cells (prior immature cells) adopted a PDX1$^{HIGH}$ phenotype in B-MAT islets, while BFP$^{HIGH}$ cells (prior mature) remained PDX1$^{HIGH}$ (Fig. 1i, j). These changes were in line with the viral transduction efficiency, which was higher in PDX1$^{LOW}$ cells (Supplementary Fig. 3a and b). While overlap in PDX1 levels in PDX1$^{LOW}$ and PDX1$^{HIGH}$ cells in B-NORM islets was observed, this likely reflects variability between experimental replicates, since the values were non-normalized. We cannot however exclude the presence of MAFA$^{LOW}$ cells that are not PDX1$^{LOW}$.

To further understand the sequence of events that occur within the islet following viral transduction, time-course experiments were performed. Notably, a shift in the normalized distribution of PDX1 fluorescence was detected beginning at 24 hrs post-infection, which persisted until 120 hrs (Supplementary Fig. 3c–f). This change was accompanied by a gradual increase in whole islet PDX1 levels (Supplementary Fig. 3g), suggesting that, at the low titers used here, immature β-cells are more susceptible to viral transduction, and that overexpression increases over time to maintain the same

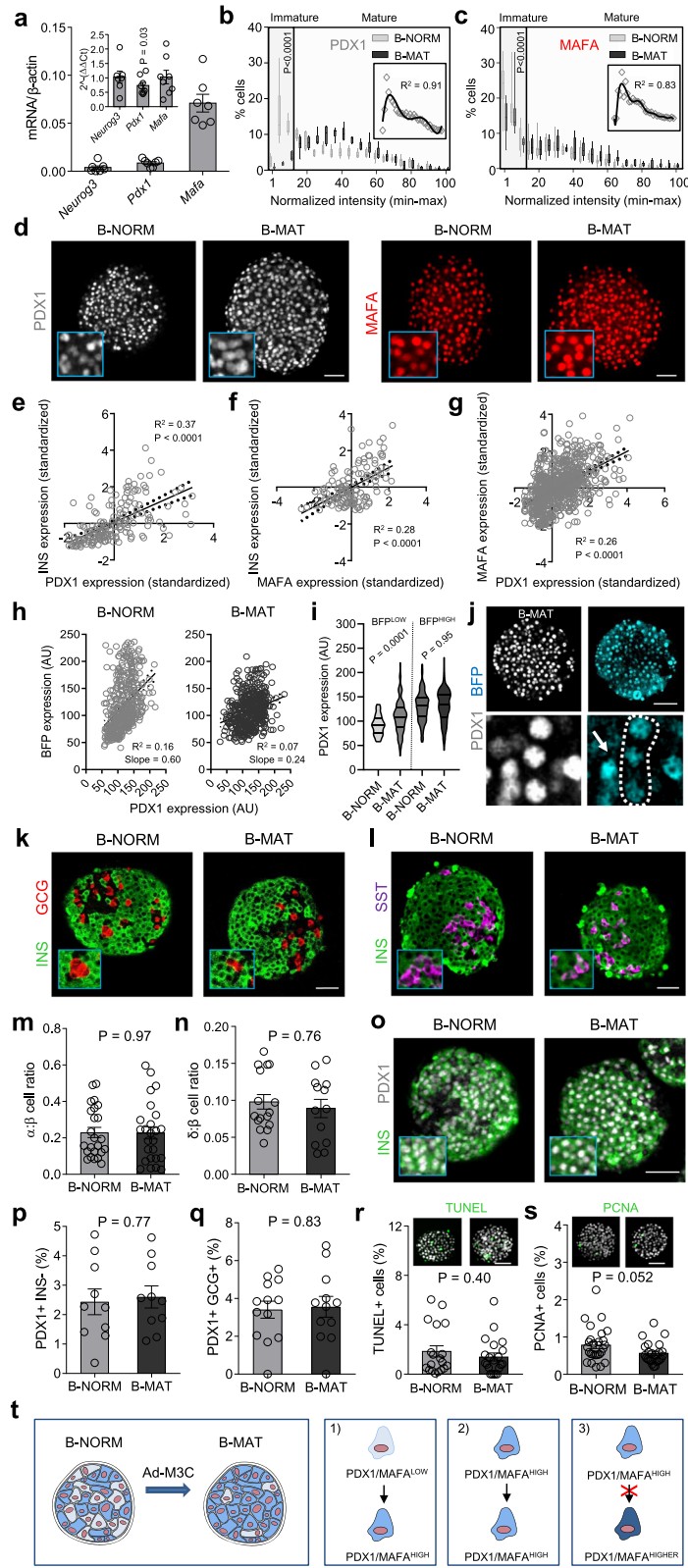

distribution. These data fit with previous reports showing that, while most β-cells are infected with adenovirus, transduction efficiency depends on the capacity of the cell to produce a protein[27]. PDX1[LOW] cells are presumably well-placed to ramp-up de novo protein synthesis, since they are also INS[LOW] (Fig. 1e) and thus unconstrained by higher rates of insulin production.

Together, these results show a shift toward proportionally more PDX[HIGH]/MAFA[HIGH] β-cells in B-MAT islets following overexpression, thus validating the model.

### α-, β- and δ-cell identity are maintained in B-MAT islets.
Further analyses of B-MAT islets detected no differences in the

**Fig. 1 Generating islets with proportionally more PDX1$^{HIGH}$/MAFA$^{HIGH}$ β-cells. a** Adenoviral *Neurog3*, *Pdx1* and *Mafa* in islets (inset, endogenous gene expression) ($n = 5$ animals; paired $t$-test). **b** Islets transduced with Ad-M3C (β-cell mature; B-MAT) lose β-cells occupying the bottom 15 percentile for PDX1 compared to controls (β-cell normal; B-NORM) (inset, non-normalized polynomial-fitted B-NORM distribution) ($n = 6$ islets/3 animals; two-way ANOVA, Bonferroni's multiple comparison) (F = 18.75, DF = 20). **c** As for **b**, but showing the frequency distribution for MAFA ($n = 8$ islets/3 animals; two-way ANOVA, Bonferonni's multiple comparison) (F = 3.03, DF = 20). **d** Images showing more homogenous PDX1/MAFA fluorescence in B-MAT islets (scale bar = 60 μm). **e–g** INS-PDX1 (**e**), INS-MAFA (**f**) and MAFA-PDX1 (**g**) are positively correlated ($n = 137$ cells, linear regression). **h** The linear correlation between PDX1 and BFP in Pdx1-BFP islets is lost following Ad-M3C transduction (B-MAT) ($n = 465$ cells/3 animals). **i** BFP$^{LOW}$ cells (prior immature) become PDX1$^{HIGH}$ in B-MAT islets, while BFP$^{HIGH}$ cells (prior mature) remain PDX1$^{HIGH}$ ($n = 93$ cells/3 animals; one-way ANOVA with Sidak's multiple comparison) (F = 52.12, DF = 3). **j** Images from Pdx1-BFP islets showing cells that underwent PDX1$^{LOW}$ -> PDX1$^{HIGH}$ conversion (arrow shows a cell that remained PDX1$^{HIGH}$) (scale bar = 50 μm) ($n = 5$ islets/3 animals; two-way ANOVA, Bonferroni's multiple comparison) (F = 2.80, DF = 18). **k–q** No differences are detected in the ratios of α- to β-cells ($n = 23$ islets/3 animals) and δ- to β-cells ($n = 18$ islets/3 animals) (**k–n**), or the proportion of PDX1 + /INS− or PDX1 + /GLU + cells ($n = 10$ islets/4 animals) (**o–q**) in B-MAT islets (unpaired t-test) (scale bar = 40 μm). **r** No difference in TUNEL+ cell numbers is detected in B-MAT islets ($n = 18$ islets/4 animals; unpaired t-test) (scale bar = 42.5 μm). **s** Cell proliferation is similar in B-NORM and B-MAT islets, as shown by PCNA staining ($n = 24$ islets/4 animals; unpaired t-test) (scale bar = 42.5 μm). **t** Transition to high PDX1/MAFA content occurs in PDX1$^{LOW}$/MAFA$^{LOW}$ cells (1), whereas PDX1$^{HIGH}$/MAFA$^{HIGH}$ cells remain unaffected (2), with PDX1/MAFA levels never surpassing those in B-NORM islets (3). Bar graphs show the mean ± SEM. Violin plot shows median and interquartile range. Box-and-whiskers plot shows median and min-max. All tests are two-sided where relevant. BFP-blue fluorescent protein; INS-insulin; GLU-glucagon; SST-somatostatin; TUNEL-terminal deoxynucleotidyl transferase dUTP nick-end labeling; PCNA-proliferating cell nuclear antigen.

ratios of α-cells or δ-cells with β-cells (Fig. 1k–n), or numbers of PDX1$^+$INS$^-$ cells (Fig. 1o, p). Expression levels of the key α-, β- and δ-cell identity markers *Arx*, *Pax6* and *Nkx6-1* (Supplementary Figure 4a), respectively, were also unaffected. Moreover, we were unable to observe differences in the numbers of PDX1$^+$ GCG$^+$ cells (Fig. 1q) or detect bihormonal cells (Supplementary Fig. 4b), consistent with the lack of viral transduction in non β-cells (Supplementary Fig. 4c). Indeed, we and others have previously shown that, at the titers used here, adenovirus is highly specific for β-cells due to reduced coxsackie virus receptor expression and low capacity for protein translation in α-cells[27–30]. However, we acknowledge that experiments using a nucleus reporter line would be needed to completely exclude transduction in α-cells. A major effect of PDX1 and MAFA overexpression on cell viability was unlikely, since no changes in expression of genes for ER stress or the unfolded protein response (UPR) were detected between B-NORM and B-MAT islets (Supplementary Fig. 4d), in line with similar ratios of TUNEL$^+$ β-cells (Fig. 1r).

Lastly, no differences in proliferation were observed between B-NORM and B-MAT islets (Fig. 1s). Thus, transduction with Ad-M3C alters the ratio of PDX1$^{LOW}$/MAFA$^{LOW}$:PDX1$^{HIGH}$/MAFA$^{HIGH}$ cells without inducing a progenitor-like state, or leading to detectable shifts in proliferation and apoptosis or the proportions of islet endocrine cell types. The schematic in Fig. 1t summarizes the loss of PDX1$^{LOW}$/MAFA$^{LOW}$ β-cell model.

**PDX1$^{LOW}$/MAFA$^{LOW}$ β-cells are transcriptionally less mature.** We next investigated whether PDX1$^{LOW}$/MAFA$^{LOW}$ cells possess a less mature transcriptional signature. Indeed, β-cell identity and function is maintained by a specific set of transcription factors, which are themselves under the control of a network of β-cell transcription factors[31] (Fig. 2a). Networks of transcription factors regulate gene expression through binding to enhancer clusters in a combinatorial manner[31]. Therefore, changes in expression of β-cell specific transcription factors impact not one, but a network of transcription factors to alter abundance of other key β-cell genes.

Analysis of published RNA-seq data showed that transcriptional levels of *MAFA* and *PDX1* are highly correlated across islet samples from 64 donors (Fig. 2b), as expected given that they belong to the same co-expression gene network module[32]. This tight correlation was also present for genes located in the same co-regulatory network such as *NEUROD1 and NKX6-1* (Fig. 2c), but not for those regulated by alternative transcriptional networks such, as *GAPDH* and *GLIS3*[31] (Fig. 2c). Similar relationships were also captured at the single cell level where human PDX$^{LOW}$

β-cells possess lower RNA abundance of genes present in the same network module, including *MAFA*, *MAFB* and *NKX6-1*[33] (Fig. 2d).

Together, these co-expression data place *PDX1* and *MAFA* at the heart of the transcription factor network that regulates β-cell identity, suggesting that the lower levels of these two key genes also indicate lower expression levels for other key β-cell transcription factors.

**Differences in PDX1 and MAFA levels sustain stimulus-secretion coupling.** Islets were subjected to detailed functional mapping to understand how differences in β-cell maturity might influence function. Multicellular Ca$^{2+}$ imaging experiments on Fluo8-loaded islets (Fig. 3a) revealed reduced Ca$^{2+}$ responses to glucose and the generic depolarizing stimulus KCl in B-MAT islets (Fig. 3b–d), which was consistent between individual islet preparations (Supplementary Fig. 5a and b). Of note, PDX1 expression levels were found to be inversely correlated with Ca$^{2+}$ amplitude in individual cells of Pdx1-BFP islets, i.e. PDX$^{LOW}$ cells tended to mount the largest Ca$^{2+}$ responses to glucose (Fig. 3a, inset). No differences in the proportion of glucose non-responsive cells were detected in B-NORM versus B-MAT islets (6.2 ± 1.7% vs 10.3 ± 2.4%, B-NORM vs B-MAT, respectively; non-significant) (Fig. 3e). These results were confirmed using the ratiometric Ca$^{2+}$ probe Fura2 (Fig. 3f–h), which again was consistent between mouse/preparation (Supplementary Fig. 5c and d). Impaired Ca$^{2+}$ fluxes in B-MAT islets were associated, but not causally-linked, with a decrease in mRNA expression of the L-type Ca$^{2+}$ channel subunits *Cacna1d* and *Cacnb2*, but not *Cacna1c* (Fig. 3i).

Suggesting a defect in electrical oscillations, Ca$^{2+}$ pulse duration was decreased in B-MAT versus B-NORM islets (Fig. 3j, k). We therefore explored if the changes in Ca$^{2+}$ fluxes observed in B-MAT islets were accompanied by defects in metabolism and amplifying signals. Using the biosensor Perceval, a ~ 2-fold decrease in glucose-stimulated ATP/ADP ratios was apparent (Fig. 3l, m). While mRNA and protein expression levels of glucokinase were not significantly different (Fig. 3n, o), we note that this does not necessarily correlate with the activity of the enzyme, which is allosterically regulated by glucokinase regulatory protein[34]. Suggestive of altered glucose-sensing, Ca$^{2+}$ and ATP/ADP glucose concentration-responses were reduced (Fig. 3p, q). Indicating impaired glucose-dependent amplifying signals, cAMP levels were decreased in response to glucose and forskolin (Fig. 3r, s). No changes in mRNA for the major murine glucose-

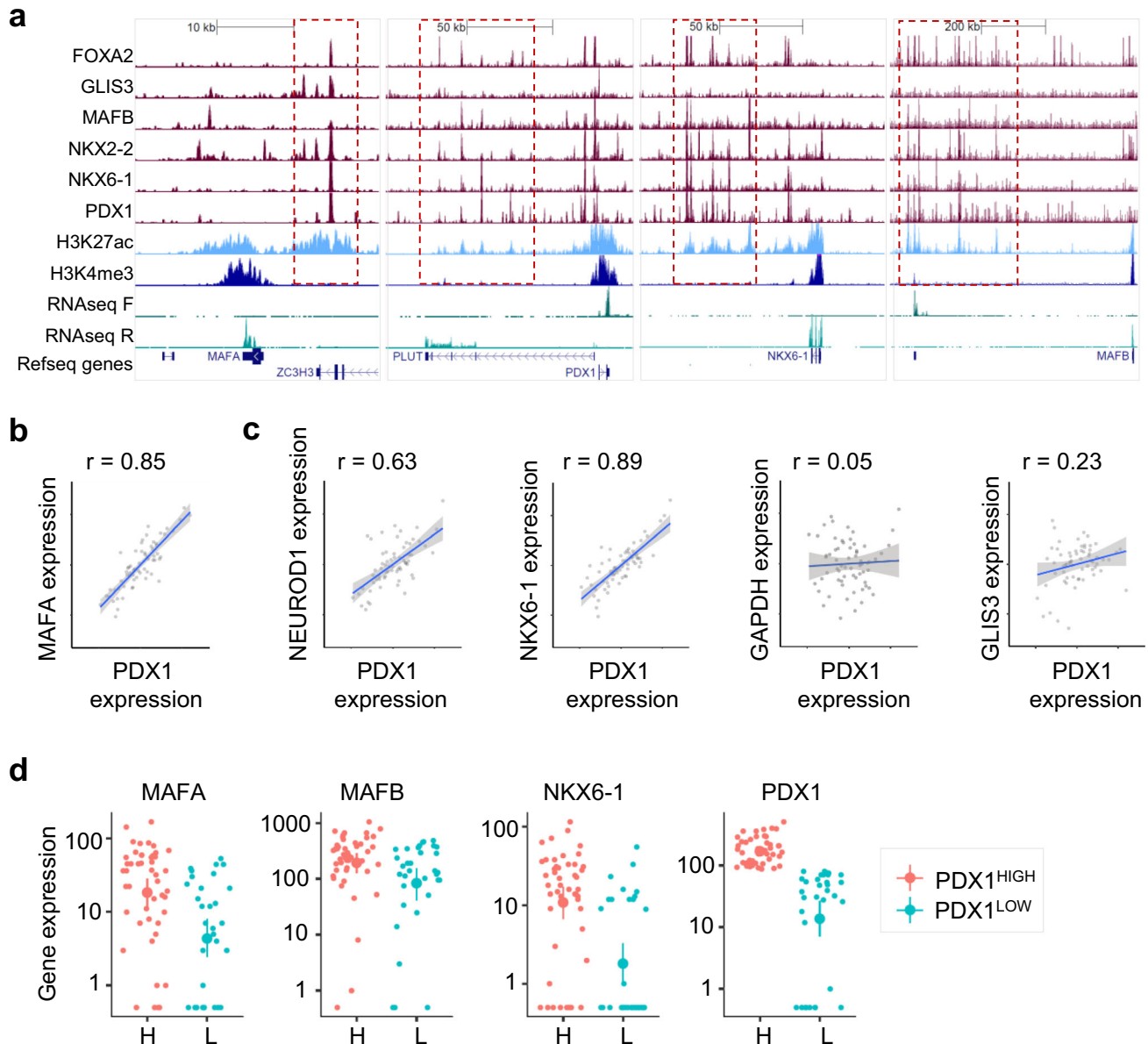

**Fig. 2 PDX1$^{LOW}$/MAFA$^{LOW}$ cells are transcriptionally immature. a** Binding of multiple transcription factors to enhancer clusters (boxed in red) regulates expression of key β-cell transcription factors in human islets. For reference, RNA-seq, H3K27ac (enhancer mark) and H3K4me3 (promoter mark) are also shown. All scales are set to 20 RPKM for ChIP-seq[32] and 20 or 60 RPKM for RNA-seq (TF strand to 60, other to 20). **b** Expression of *MAFA* and *PDX1* correlate over 64 human islet samples. The axes represent normalized expression values (−3 to 3) for each gene used for the co-expression network analysis[31]. **c** Correlation of expression of mRNA for *PDX1* and *NEUROD1*, *NKX6-1*, *GAPDH* and *GLIS3* across 64 human islet samples. The axes represent normalized expression values (−3 to 3) for each gene used for the co-expression network analysis[31]. **d** Single cell gene expression levels for *MAFA*, *MAFB*, *NKX6-1* and *PDX1* in cells with high and low mRNA levels for PDX1. Analysis was performed using Monocle, the y-axis representing gene expression levels in log10 scale. Datasets were obtained from[31,33].

regulated adenylate cyclase, *Adcy8*[35], were detected (Fig. 3t). Potentially unifying the abovementioned metabolic and electrical observations, analysis of PDX1 and Ca$^{2+}$ targets in B-MAT islets revealed changes in expression of both *G6pc2* and *Ascl1*[36,37] (Fig. 3u).

Thus, islets with proportionally more PDX1$^{HIGH}$/MAFA$^{HIGH}$ β-cells display profound defects in metabolism and stimulus-secretion coupling, including ionic and amplifying signals.

**Differences in PDX1 and MAFA levels sustain islet dynamics and insulin secretion.** We investigated whether a reduction in PDX1$^{LOW}$/MAFA$^{LOW}$ cells would lead to a decline in

functional subpopulations shown to drive islet dynamics, some of which possess an immature or energetic phenotype. Fast Ca$^{2+}$ recordings (20 Hz) detected cells whose activity overlapped that of the rest of the population. These cells, algorithmically-identified as 'hubs', comprise ~ 1–10% of the β-cell population, orchestrate islet responses to glucose and show immature traits (PDX1$^{LOW}$, NKX6-1$^{LOW}$, INS$^{LOW}$)[21]. The proportion of hubs was decreased in B-MAT islets (Fig. 4a), most likely due to a reduction in the number of immature cells able to act as hubs combined with decreased expression of *Gjd2* (Fig. 4b), which encodes the gap junction protein connexin 36 (Cx36). The loss of hubs was associated with a reduction in indices of coordinated β-cell activity ('connectivity') (Fig. 4c), typified by a shift toward

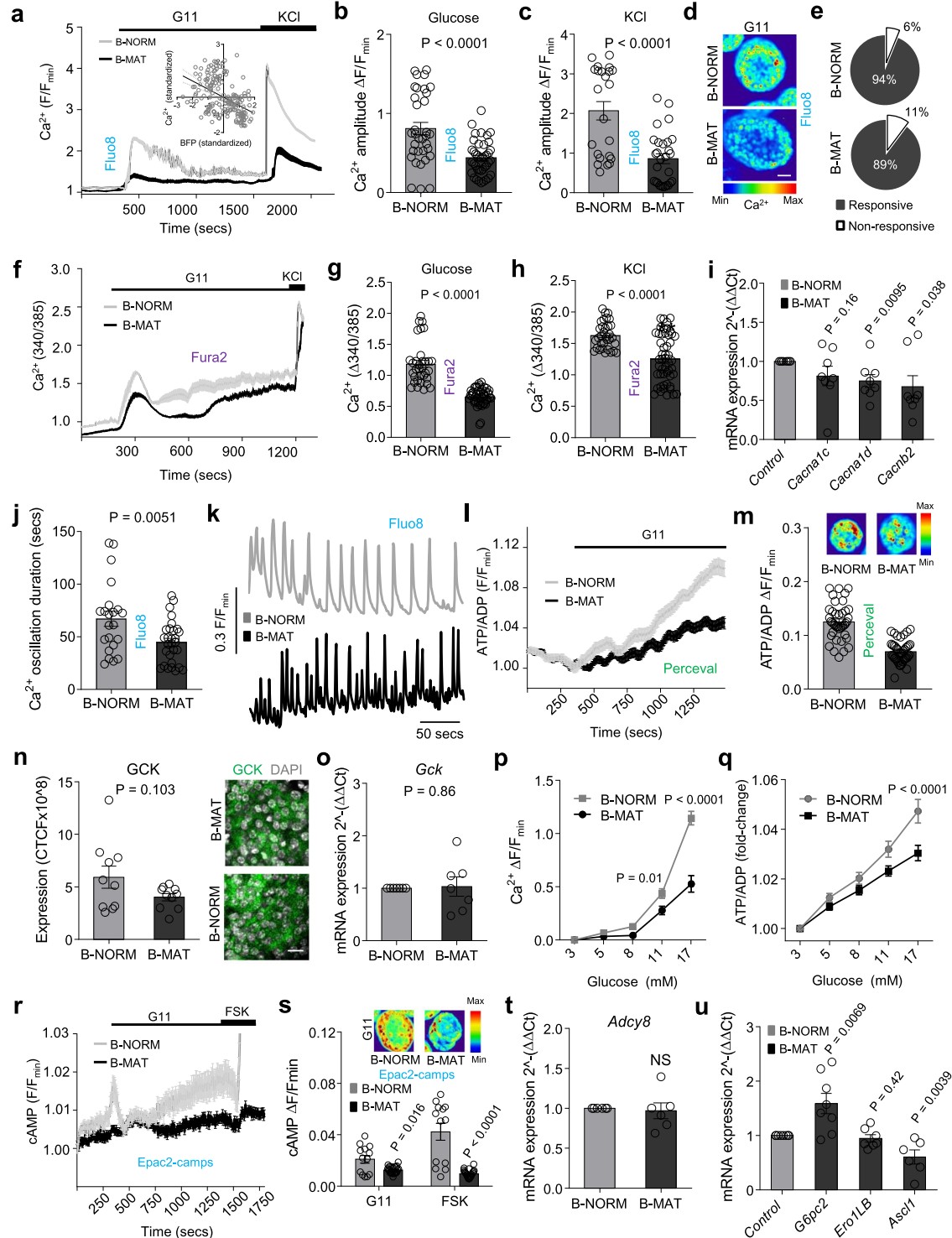

more stochastic β-cell population responses (Fig. 4d, e) (Supplementary Movies 1 and 2).

As predicted from the impairments in $Ca^{2+}$ fluxes, metabolism, amplifying signals and β-cell-β-cell connectivity, glucose- and Exendin-4-stimulated insulin release was markedly decreased in B-MAT islets (Fig. 4f, g), despite an increase in insulin content (Fig. 4h). Insulin secretion was similar in B-NORM and B-MAT islets when uncorrected for content, suggesting that B-MAT islets release only a fraction of their secretory granule pool in response to glucose (Supplementary Fig. 5e–f). However, fold-change

insulin secretion remained significantly decreased in B-MAT islets (Supplementary Fig. 5g). Super-resolution imaging revealed no differences in insulin granule density at the membrane (Fig. 4i), in line with unchanged expression of mRNA for the exocytotic machinery (e.g., *Stx1a*, *Snap25* and *Vamp2*) (Fig. 4j). Implying the presence of normal insulin gene regulation, *Ins1* and *Ins2* mRNA levels were unaffected (Fig. 4k, l). The loss of incretin-responsiveness was surprising given that gut- and islet-derived GLP1[38,39] potently upregulates the sensitivity of insulin granules for exocytosis[38]. Further analyses showed a large

**Fig. 3 Differences in PDX1 and MAFA levels contribute to islet signaling. a–c** $Ca^{2+}$ fluxes (**a**) in response to glucose (**b**) or glucose + KCl (**c**) are impaired in B-MAT islets, also shown by representative images (**d**) ($n = 34$ islets/4 animals; unpaired t-test) (scale bar = 40 μm). Inset in (**a**) shows an inverse correlation between glucose-stimulated $Ca^{2+}$ amplitude and BFP expression in individual β-cells (Pdx1-BFP; $n = 6$ islets/3 animals; $R^2 = 0.21$, $P < 0.0001$) (G11, 11 mM glucose; KCl, 10 mM). **e** No differences in the % glucose-responsive β-cells are detected in B-MAT islets ($n = 34$ islets/4 animals; unpaired t-test). **f–h** As for (**a–c**), but using Fura2 ($n = 33$ islets/4 animals; unpaired t-test). **i** Expression of genes encoding CACNA1D and CACNB2 $Ca^{2+}$ channel subunits is reduced in B-MAT islets ($n = 8$ animals; paired t-test). **j, k** $Ca^{2+}$ pulse duration is reduced in B-MAT islets, as shown by summary bar graph (**j**) and traces (**k**) ($n = 8$ islets/4 animals; unpaired $t$-test). **l, m** ATP/ADP ratios are reduced in B-MAT islets, as shown by mean traces (**l**) and summary bar graph (**m**) ($n = 40$ islets/4 animals; unpaired $t$-test). **n, o** GCK expression (**n**) tends to be reduced in B-MAT islets ($n = 10$ islets/2 animals; paired t-test), although *Gck* levels are normal (**o**) ($n = 7$ animals; paired t-test) (scale bar = 15 μm). **p, q** $Ca^{2+}$ (**p**) and ATP/ADP (**q**) responses to increasing glucose concentration are decreased in B-MAT islets ($Ca^{2+}$: $n = 11$ islets/5 animals; two-way ANOVA F = 20.36, DF = 4) (ATP/ADP: $n = 37$ islets/5 animals, two-way ANOVA; F = 6.10, DF = 4) (Bonferroni's multiple comparison). **r, s** Mean traces (**r**) and bar graph (**s**) showing reduced cAMP levels in response to glucose and forskolin (FSK, 100 μM) in B-MAT islets ($n = 13$ islets; unpaired t-test). **t** *Adcy8* expression remains unchanged in B-MAT islets ($n = 6$ animals; paired t-test). **u** *G6pc2* and *Ascl1* are up- and down-regulated, respectively, in B-MAT islets ($n = 6$ animals; paired t-test). Color scale shows $Ca^{2+}$ as min (0%) to max (100%) value. Bar graphs and traces show the mean ± SEM. All tests are two-sided where relevant. CTCF-corrected total cell fluorescence.

decrease in glucagon-like peptide-1 receptor (GLP1R) mRNA and protein expression (Fig. 4m, n), which was accompanied by impairments in Exendin-4-stimulated cAMP (Fig. 4o–q) and $Ca^{2+}$ (Fig. 4r, s) signals.

As such, differences in PDX1 and MAFA levels contribute to islet $Ca^{2+}$ dynamics and insulin release.

**A balance between PDX1$^{LOW}$ and PDX1$^{HIGH}$ β-cells is required for human islet function.** We next examined whether differences in maturity status of individual β-cells might represent a conserved route for islet function in human islets. As expected, transduction with Ad-M3C (β human mature; B-hMAT) led to increases in exogenous *Neurog3, Pdx1* and *Mafa* mRNA levels (Fig. 5a). Endogenous levels of *NEUROG3, MAFA* and *PDX1* were unchanged (Fig. 5b).

PDX1 fluorescence intensity distribution, visualized using antibodies with cross-reactivity against both human and mouse protein, was bimodal in B-hNORM (β human normal) islets, with peaks corresponding to PDX1$^{LOW}$ and PDX1$^{HIGH}$ populations (Fig. 5c, d), again arbitrarily defined by a 15% cut-off. A similar distribution of PDX1 fluorescence was detected when normalized to DAPI staining (Supplementary Fig. 6a and b), or when only PDX1 + /INS1 + cells were considered (Supplementary Fig. 6c). The number of cells occupying the PDX1$^{LOW}$ range (i.e. immature) was decreased in B-hMAT compared to B-hNORM islets (Fig. 5c, d), suggesting a shift toward a more homogenous distribution of β-cell maturity. As for mouse islets, PDX1 and INS expression were found to be correlated (Fig. 5e). We were unable to extend findings to MAFA and NEUROG3, since attempts at antibody staining were unsuccessful in the isolated islet.

In any case, B-hMAT islets presented with reductions in $Ca^{2+}$ responses to glucose or glucose + KCl (Fig. 5f–i), without alterations in the proportion of responsive cells (Fig. 5j), recorded using the genetically-encoded $Ca^{2+}$ indicator, GCaMP6. These defects in $Ca^{2+}$ fluxes were associated with significantly lowered expression of mRNA for the L and T-type $Ca^{2+}$ channel subunits *CACNA1C, CACNA1D* and *CACNA1G*, as well as the $Na^+$ channel subunits *SCN1B, SCN3A* and *SCN8A* (Fig. 5k). The disruption of islet $Ca^{2+}$ dynamics in B-hMAT islets was accompanied by decreases in gap junction protein expression (Fig. 5l), proportion of hub cells (Fig. 5m) and β-cell-β-cell coordination (Fig. 5n, o). Although glucose-stimulated insulin secretion was similar in B-hMAT and B-hNORM islets (Fig. 5p), the former released only a fraction of their granules when corrected for the increase in total insulin (Fig. 5q, r). Thus, altering the ratio of PDX1$^{LOW}$/MAFA$^{LOW}$:PDX1$^{HIGH}$/MAFA-$^{HIGH}$ cells has similar effects on mouse and human islet function (Fig. 5s).

**Increases in the proportion of PDX1$^{LOW}$/MAFA$^{LOW}$ β-cells impair islet function.** To investigate whether a balance between mature and immature β-cells is required for normal islet operation, the opposite model was generated by inducing a higher proportion of PDX$^{LOW}$ cells across the population. Application of short hairpin RNAs against *Pdx1* resulted in a left-shift in the distribution of PDX1 protein fluorescence intensities, indicative of a decrease in the number of PDX1$^{HIGH}$ β-cells (Fig. 6a, b), in line with downregulation of *Pdx1* mRNA (Fig. 6c). MAFA protein fluorescence intensity was also decreased (Fig. 6a, b), supporting the RNA-seq analysis showing that PDX1 and MAFA belong to the same regulatory network. Immunofluorescence analyses showed no changes in the α- to β-cell ratio, indicating that β-cells were unlikely to be de-differentiating toward an α-cell phenotype (Fig. 6d). Islets with proportionally more PDX1$^{LOW}$ β-cells (B-IMMAT) presented with lowered insulin content (Fig. 6e), a tendency toward increased basal hormone levels (Fig. 6f), and absence of glucose-stimulated insulin release that could be restored using Exendin-4 (Fig. 6f, g). Similar to overexpression experiments, glucose- and KCl-stimulated $Ca^{2+}$ fluxes were impaired (Fig. 6h–j), together with decreased expression of the VDCC subunits *Cacna1d* and *Cacnb2* (but not *Cacna1c*) (Fig. 6k).

Together, these experiments demonstrate that increasing the proportion of either PDX$^{LOW}$/MAFA$^{LOW}$ or PDX$^{HIGH}$/MAFA-$^{HIGH}$ β-cells results in a similar islet phenotype (i.e., perturbed insulin secretion, ionic fluxes and β-cell population dynamics) (Fig. 6l).

**Differences in PDX1 and MAFA expression are encoded by the islet context.** Since regulated $Ca^{2+}$ fluxes are critical for maintaining β-cell differentiation[37], we wondered whether PDX$^{LOW}$/MAFA$^{LOW}$ or PDX$^{HIGH}$/MAFA$^{HIGH}$ β-cells might help maintain their own phenotype in the islet setting due to differences in their $Ca^{2+}$ signals (i.e. through a feedforward mechanism). To test this, we repeated immunohistochemical analyses in dissociated β-cells where cell-cell communications are disrupted, and $Ca^{2+}$ dynamics are less pronounced and more stochastic[21]. Unexpectedly, the PDX1 and MAFA intensity distributions were right-shifted in dissociated islets, with β-cells in the PDX1$^{LOW}$ and MAFA$^{LOW}$ range no longer apparent after 24 hr culture (Fig. 7a–c). A PDX1$^{LOW}$ subpopulation could still be detected 3 h after coverslip attachment (Fig. 7d), and PDX1 frequency distribution was similar in scrRNA or sh*GJD2*-treated islets (Fig. 7e, f). As such, β-cells likely undergo a gradual adjustment in maturity status following dissociation rather than apoptosis/cell death, these changes occur independently of changes in gap junction signaling (e.g. due to alterations in paracrine input), and

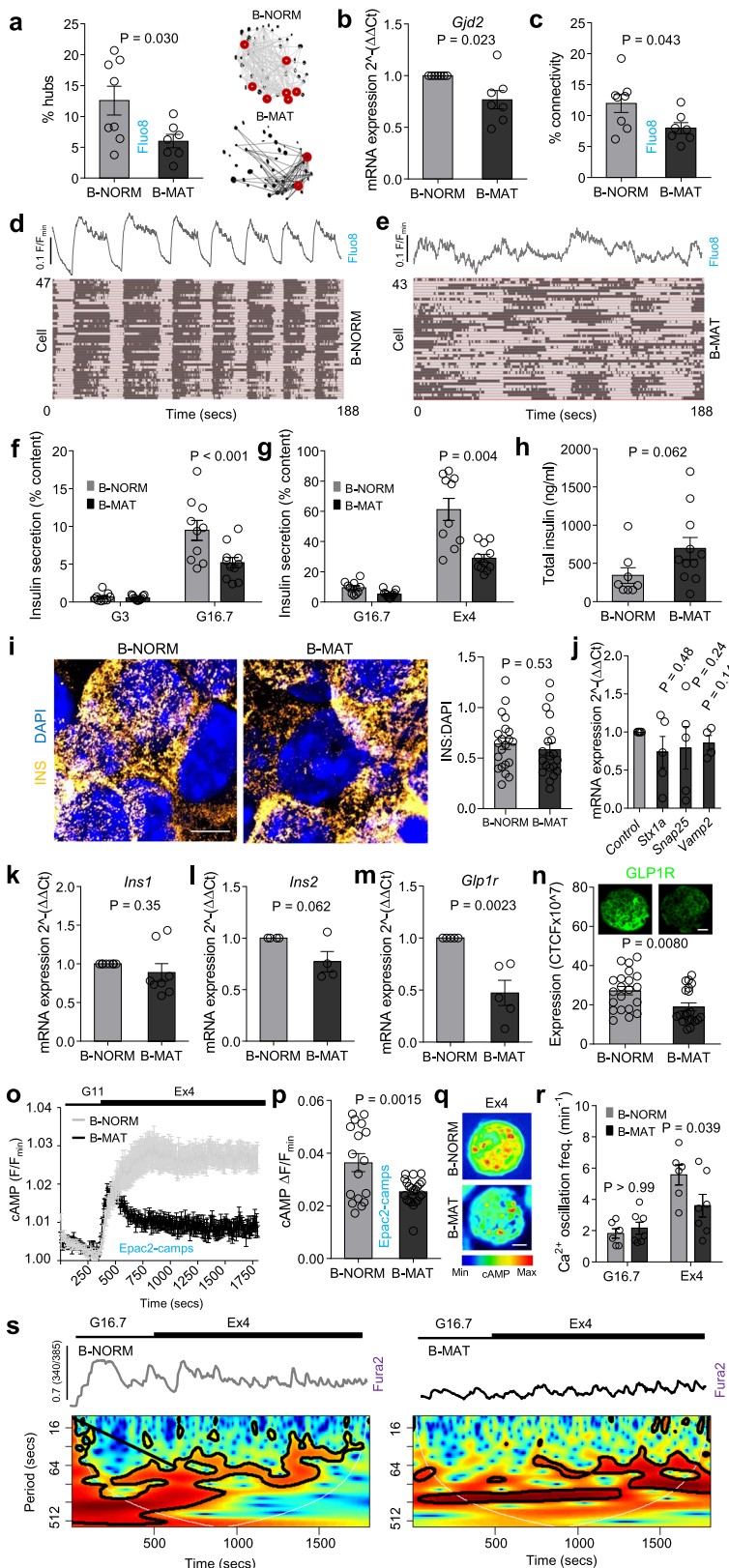

should be considered when extrapolating results from studies in dissociated β-cells.

To further investigate whether Ca²⁺ dynamics might contribute to β-cell maturity directly in the islet setting, we turned to a chemogenetic strategy to precisely control membrane potential. Conditional β-cell silencing was achieved using Ins1Cre animals

crossed to a strain harboring stop-floxed alleles for hM4Di, a mutant muscarinic receptor with low affinity for endogenous acetylcholine[40]. Upon administration of designer ligand, the G_i pathway is activated specifically in β-cells, leading to long-lasting electrical silencing via effects on cAMP and G protein-coupled inwardly-rectifying potassium channels[40,41]. We used this

**Fig. 4 Both PDX1$^{LOW}$/MAFA$^{LOW}$ and PDX1$^{HIGH}$/MAFA$^{HIGH}$ β-cells are required for islet dynamics and insulin secretion. a–c** Hub cell proportion (red circles) (**a**) ($n = 7$ islets/3 animals; unpaired $t$-test), mRNA for *Gjd2* (**b**) ($n = 7$ animals; paired $t$-test) and coordinated β-cell activity (connectivity) (**c**) ($n = 7$ islets/3 animals; unpaired $t$-test) are all decreased in B-MAT islets. **d, e** Raster plots showing β-cell activity profiles in B-NORM (**d**) and B-MAT islets (**e**). **f–h** Loss of PDX$^{LOW}$/MAFA$^{LOW}$ β-cells leads to reductions in glucose (**f**)- and Exendin-4 (**g**)-stimulated insulin secretion ($n = 10$ replicates/4 animals; two-way ANOVA, Bonferroni's multiple comparison) (G16.7: F = 7.89, DF = 1) (Ex4: F = 13.40, DF = 1), despite an increase in insulin content (**h**) ($n = 8$ replicates/4 animals; Mann–Whitney $U$-test). Samples were run together, but due to the relative magnitude of the Exendin-4 response, results are displayed separately (G3, 3 mM glucose; G16.7, 16.7 mM glucose; Ex4, 20 nM Exendin-4). **i** Images and summary bar graph showing insulin granule density at the membrane in B-NORM and B-MAT islets (scale bar = 6 μm) ($n = 12$ islets/6 animals; unpaired $t$-test). **j** No differences in *Stx1a*, *Snap25* and *Vamp2* expression are detected in B-MAT islets ($n = 5$ animals; paired $t$-test). **k, l** *Ins1* (**k**) and *Ins2* (**l**) levels are unchanged in B-MAT islets ($n = 4$ animals; paired $t$-test). **m, n** GLP1R mRNA (**m**) ($n = 5$ animals, paired $t$-test) and protein (**n**) ($n = 20$ islets/4 animals, unpaired $t$-test) expression are reduced in B-MAT islets (scale bar = 25 μm). **o–q** Maximal Exendin-4-stimulated cAMP rises are blunted in B-MAT islets, shown by mean traces (**o**) and summary bar graph (**p**), as well as representative images (scale bar = 25 μm) (**q**) ($n = 17$ islets/2 animals; unpaired $t$-test) (G11, 11 mM glucose; Ex4, 20 nM Exendin-4). **r, s** Exendin-4-stimulated Ca$^{2+}$-spiking is blunted in B-MAT islets (**r**), confirmed using wavelet analysis (**s**) (mean wave shown) ($n = 6$ islets/3 animals; two-way ANOVA, Bonferroni's multiple comparison) (F = 4.40, DF = 1) (G16.7, 16.7 mM glucose; Ex4, 20 nM Exendin-4). Bar graphs and traces show the mean ± SEM. All tests are two-sided where relevant.

manoeuvre to generate D-NORM and D-MAT islets, which possess wild-type (control) or hM4Di alleles, respectively.

Specific expression of hM4Di in β-cells was confirmed via expression of a Citrine reporter (Fig. 7g). We first tested hM4Di functionality using the second-generation hM4Di agonist J60. As expected, J60 silenced β-cell Ca$^{2+}$ spiking activity within 15 mins of application in D-MAT but not D-NORM islets (Fig. 7h, i) (Supplementary Movies 3 and 4). No inhibitory effects of hM4Di alone were detected, with a small but significant increase in basal Ca$^{2+}$ levels detected in the presence of the receptor (Fig. 7j). By contrast to J60, the first-generation agonist clozapine N-oxide (CNO) decreased Ca$^{2+}$ levels (Fig. 7j) and Ca$^{2+}$ oscillation frequency (Fig. 7k, l) after 3 h, but did not completely suppress β-cell activity. We took advantage of this property to disrupt rather than ablate the β-cell Ca$^{2+}$ signaling network.

Following treatment of islets with CNO for 48 h, immunostaining showed a decrease in the number of cells in the lowest PDX1 and MAFA fluorescence intensity bins in D-MAT islets (Fig. 7m–o). While washout of CNO for 2 hrs restored baseline Ca$^{2+}$ levels in D-MAT islets (Fig. 7p), Ca$^{2+}$ responses to both glucose and KCl remained markedly impaired (Fig. 7q, r), closely resembling those seen in both B-MAT and B-IMMAT islets. Furthermore, chemogenetic disruption decreased the proportion of cell-cell connectivity and hubs (Fig. 7s, t), which was associated with a shift to more stochastic islet dynamics (Supplementary Movies 5 and 6), as expected. Gene expression analyses in D-MAT islets showed significant reductions in *Cacna1d* and *Gjd2*, with *Cacna1c*, *Cacnb2*, *Ins1*, *Ins2*, *Glp1r* and *Gck* all remaining similar to D-NORM controls (Supplementary Fig. 7). The phenotype of D-MAT islets was unlikely to be dependent on insulin signaling (or loss thereof), since application of insulin receptor antagonist S961 to wild-type islets increased the proportion of PDX$^{LOW}$ rather than PDX1$^{HIGH}$ β-cells (Supplementary Fig. 6d and e). Moreover, selection by cell death was unlikely to feature in D-MAT islets, since a reduction (but not ablation) in Ca$^{2+}$ signaling would be expected to alleviate cell stress[42].

These chemogenetic experiments suggest that either: 1) differences in PDX1 and MAFA expression are maintained via Ca$^{2+}$ signaling patterns encoded by the islet context; or 2) PDX$^{LOW}$/MAFA$^{LOW}$ β-cells within the islet represent a ER-stressed or transitory subpopulation, which recovers its identity when rested[42,43] (Fig. 7u).

**The balance between PDX1$^{LOW}$ and PDX1$^{HIGH}$ β-cells influences downstream gene expression**. To define the transcriptional profile of islets containing proportionally more PDX1$^{HIGH}$ β-cells, we performed differential gene expression analysis (DGE)

on control and B-MAT mouse islets. To increase PDX1 and MAFA levels throughout the islet, we developed a doxycycline-inducible mouse model for the cistronic expression of PDX1, MAFA and NEUROG3. The model was generated by crossing RIP7rtTA mice with those harboring NEUROG3/PDX1/MAFA/mCherry under the control of a tetracycline response element (Tet-MAT) (Fig. 8a). As expected, Tet-MAT islets displayed increased expression of *Pdx1, Mafa* and *Neurog3* in comparison to control islets (Tet-NORM) (Fig. 8b). This was accompanied by a decrease in the number of PDX1$^{LOW}$ β-cells (Fig. 8c, d), as well as impaired Ca$^{2+}$ fluxes (Fig. 8e–g), without evidence of a generalized PDX1 overexpression (fluorescence intensity = 11044 ± 1837 AU versus 12679 ± 1813 AU, Tet-NORM versus Tet-MAT, respectively; non-significant), Thus, we were able to confirm results in a third independent model, further demonstrating the robustness of the adenoviral transduction model.

Doxycycline-treated islets from Tet-NORM and Tet-MAT mice were then subjected to transcriptomic profiling using RNA-seq. Differential gene expression analysis (DGE) revealed 83 genes whose expression was significantly altered between Tet-NORM and Tet-MAT islets (at adjusted $p$-value < 0.05) (Fig. 8h). The majority (94%) of these genes were upregulated in Tet-MAT islets (Fig. 8h). Gene annotation analysis (DAVID)[44] revealed that significantly upregulated genes were enriched for gene ontology clusters related to β-cell function and identity (Fig. 8i, j), confirming the validity of the model at the transcriptomic level. Gene set enrichment analysis (GSEA) also revealed upregulation of other molecular pathways such as metabolic processes linked to glucose and carbohydrate derivatives (Fig. 8k). Closer inspection of the significantly upregulated genes revealed a number of candidates that might impact insulin secretion including *Ucn3*, *G6pc2*, *Cox6a2*, *Rgs4* and *Pkib*[45–49], confirmed using RT-qPCR (Fig. 8l). Taken together, these results show that increasing the proportion of PDX1$^{HIGH}$ β-cells in the islet leads to upregulation of key β-cell identity markers, but also results in differential regulation of pathways, such as those involved in cellular nutrient metabolism.

**Restoring the balance between PDX1$^{LOW}$ and PDX1$^{HIGH}$ β-cells is protective**. A decrease in the expression of β-cell identity markers such as NKX6-1, PDX1 and MAFA occurs during metabolic stress[50,51]. This may alter the balance between immature and mature β-cells, with consequences for normal islet function. We therefore examined whether restoring the balance between PDX$^{LOW}$ or PDX$^{HIGH}$ β-cells would prevent islet failure in response to lipotoxic insult.

Islets treated for 48 h with high concentration of the fatty acid palmitate showed a left-shift in the PDX1 fluorescence intensity

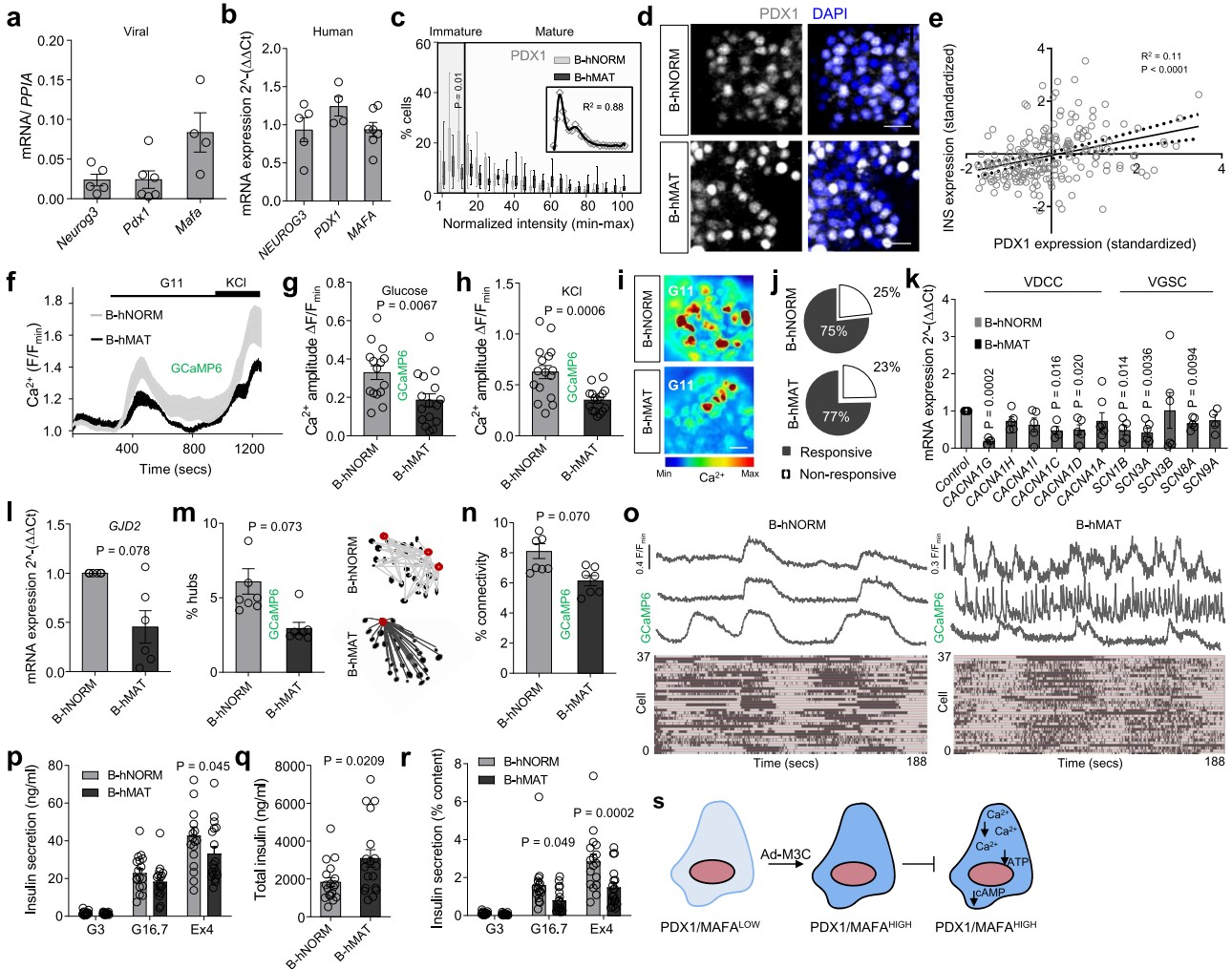

**Fig. 5 Both PDX1^LOW and PDX1^HIGH β-cells contribute to human islet function. a**, **b** Ad-M3C increases *Neurog3*, *Pdx1* and *MafA* expression (**a**), while no differences are detected in native *NEUROG3*, *PDX1* and *MAFA* expression (**b**) ($n = 4$–8 donors). **c** Ad-M3C increases the proportion of cells expressing high PDX1 levels (B-hMAT) (inset is the non-normalized B-hNORM distribution fitted with a polynomial) ($n = 13$ islets/4 donors; two-way ANOVA, Bonferroni's multiple comparison) (F = 4.14, DF = 20). **d** Representative images showing loss of PDX1^LOW cells in B-hMAT islets (detected using a PDX1 antibody with reactivity against mouse and human protein) (scale bar = 42.5 μm). **e** PDX1 and INS1 are positively correlated in individual cells from B-hNORM islets ($n = 220$ cells). **f**–**h** Ca$^{2+}$ traces (**f**) showing decreased responsiveness to glucose (**g**) and KCl (**h**) in B-hMAT islets ($n = 16$ islets/3 donors; unpaired *t*-test). **i**, **j** as for (**f**–**h**), but representative images (scale bar = 25 μm) showing loss of glucose-stimulated Ca$^{2+}$ rises in B-hMAT but not B-hNORM islets (**i**), despite no differences in the proportion of responsive cells (**j**) ($n = 16$ islets/3 donors; unpaired *t*-test). **k** The VDCC and Na$^+$ channel subunits *CACNA1G*, *CACNA1C*, *CACNA1D*, *SCN1B*, *SCN3A* and *SCN8A* are all downregulated in B-hMAT islets ($n = 4$–6 donors; paired *t*-test). **l**–**o** *GJD2* expression (**l**) is decreased in B-hMAT islets ($n = 6$ donors; paired t-test), which is associated with a decrease in the number of hubs (circled in red) (**m**) and coordinated β-cell-β-cell activity (connectivity) (**n** and **o**) (representative traces are from 'connected' cells; raster plots show intensity over time) ($n = 7$–8 islets/3 donors; unpaired *t*-test). **p**–**r** Non-normalized Insulin secretion is similar in B-hMAT and B-hNORM islets (**p**), although B-hMAT islets only release a fraction of their total insulin (**q** and **r**) (% insulin content = secreted insulin / total insulin) ($n = 17$–18 replicates/5 donors; unpaired *t*-test and two-way ANOVA, Bonferroni's multiple comparison). **s** Schematic showing proposed changes occurring in β-cells in B-hMAT islets. Bar graphs and traces show the mean ± SEM. Box-and-whiskers plot shows median and min-max. All tests are two-sided where relevant. Color scale shows Ca$^{2+}$ as min (0%) to max (100%) value. GCaMP6-genetically-encoded Ca$^{2+}$ indicator; VDCC-voltage-dependent Ca$^{2+}$ channels; VGSC-voltage-gated Na$^+$ channels; *GJD2*-Gap junction delta-2 protein encoding Connexin-36.

distribution, primarily due to a decrease in the number of PDX1^HIGH β-cells (Fig. 9a). Transduction with Ad-M3C reversed this decrease (Fig. 9b, c), with PDX1 expression levels being indistinguishable from BSA controls. Functional assessment of palmitate-treated islets revealed ~50% lowered Ca$^{2+}$ fluxes in response to both glucose and KCl (Fig. 9d–f). Pertinently, these deficits could be prevented using Ad-M3C (Fig. 9d–f). From this, it can be inferred that re-establishing the balance between PDX1^LOW and PDX1^HIGH β-cells, and thus restoring differences

in PDX1 levels, protects against islet failure during metabolic stress.

## Discussion

It is becoming increasingly apparent that β-cells can be grouped into subpopulations according to their transcriptomic and protein signatures. In particular, the existence of immature β-cells in the normal adult islet poses a conundrum, since this subpopulation is generally considered to be poorly functional when viewed in

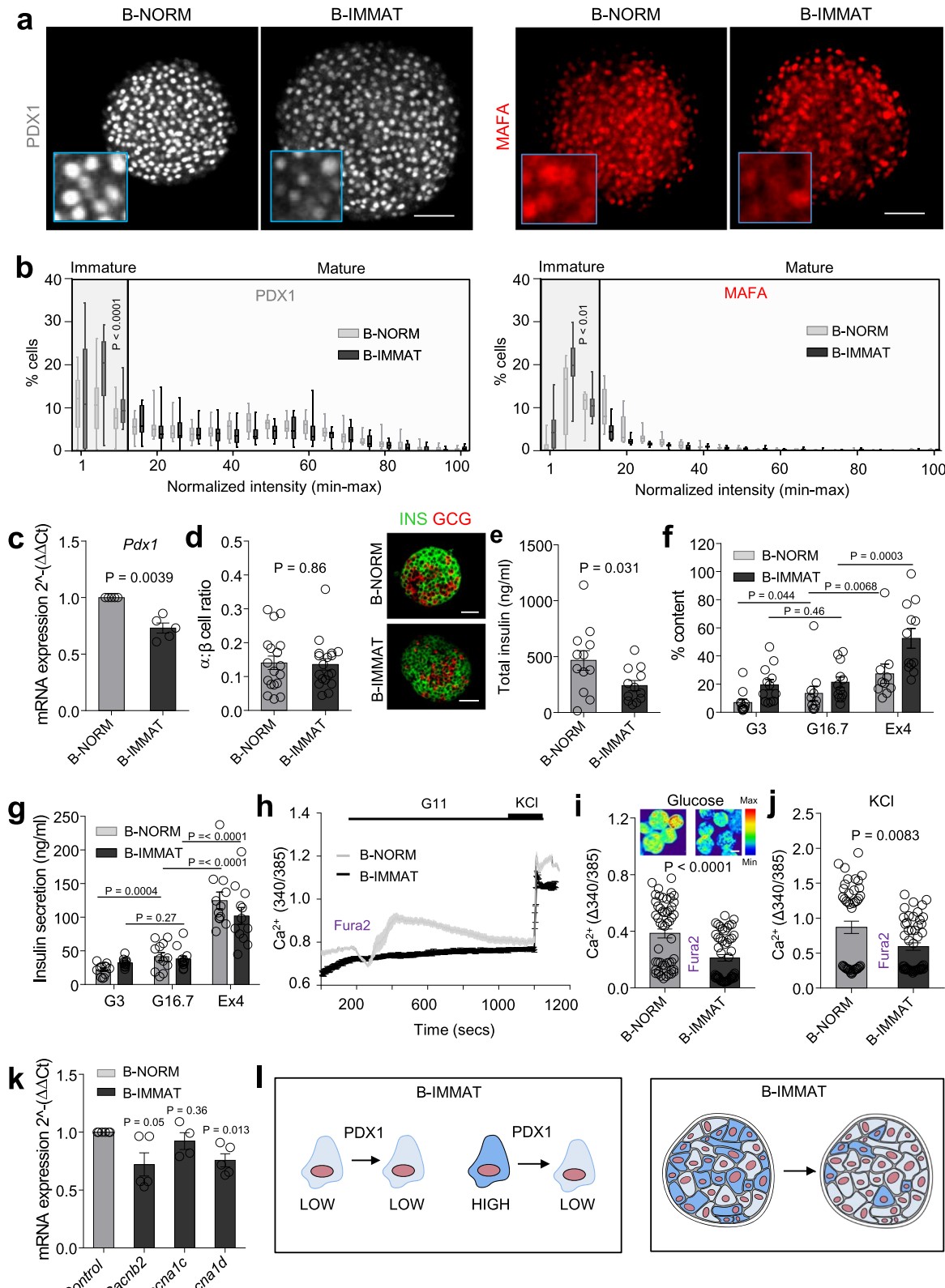

isolation[4,7,8,14]. Despite this, no previous studies have imposed changes on β-cell maturity while examining functional outcomes. Using multiple models, we show here that differences in β-cell maturity, arbitrarily defined using PDX1 and MAFA levels, are needed across the population for proper islet function. An increase in the proportion of mature (PDX1$^{HIGH}$/MAFA$^{HIGH}$) β-cells is associated with islet failure due to impaired ionic fluxes, metabolism and cell-cell connectivity (schematic in Fig. 9g). Furthermore, redressing the balance between immature (PDX1$^{LOW}$/MAFA$^{LOW}$) and mature (PDX1$^{HIGH}$/MAFA$^{HIGH}$) β-cells restores islet function under conditions of metabolic stress. Thus, our studies provide evidence that both PDX1$^{LOW}$/MAFA$^{LOW}$ and

**Fig. 6 A proportional increase in PDX1^LOW/MAFA^LOW β-cells impairs islet function. a** sh*Pdx1* increases the proportion of β-cells in the islet with low levels of PDX1 and MAFA (β-cell immature; B-IMMAT) (scale bar = 60 μm). **b** Quantification of PDX1 and MAFA expression intensity shows an increase in β-cells occupying the bottom 15 percentile in B-IMMAT islets (*n* = 13–14 islets/3 animals; two-way ANOVA, Bonferroni's multiple comparison) (PDX1: F = 2.38, DF = 20) (MAFA: F = 3.20, DF = 20). **c** RT-qPCR showing a decrease in *Pdx1* expression levels in B-IMMAT islets (*n* = 5; paired *t*-test). **d** Induction of homogenous β-cell immaturity does not alter the α- to β-cell ratio (scale bar = 42.5 μm) (*n* = 18 islets/ 2–3 animals; unpaired *t*-test). **e–g** B-IMMAT islets display decreased insulin content (**e**), increased basal insulin release and absence of significant glucose-stimulated insulin secretion (**f** and **g**) (*n* = 10–12 replicates/4 animals; paired *t*-test and one-way ANOVA, Sidak's multiple comparison) (G3, 3 mM glucose; G16.7, 16.7 mM glucose; Ex4, 20 nM Exendin-4). **h–j** Ca^2+ traces (**h**) and bar graphs (**i** and **j**) showing impaired responses to glucose and glucose + KCl in B-IMMAT islets (*n* = 49–51 islets/ 4–5 animals; unpaired *t*-test) (representative images shown above bar graph, scale bar = 75 μm). **k** mRNA for the L-type VDCC subunits *Cacnb2* and *Cacna1d* are significantly downregulated in B-IMMAT islets (*n* = 5–6; paired *t*-test). **l** Schematic showing the proposed changes in B-IMMAT islets. Color scale shows Ca^2+ as min (0%) to max (100%) value. Bar graphs and traces show the mean ± SEM. Box-and-whiskers plot shows median and min-max. All tests are two-sided where relevant. sh*Pdx1*- short hairpin RNA against *Pdx1*; VDCC-voltage-dependent Ca^2+ channels.

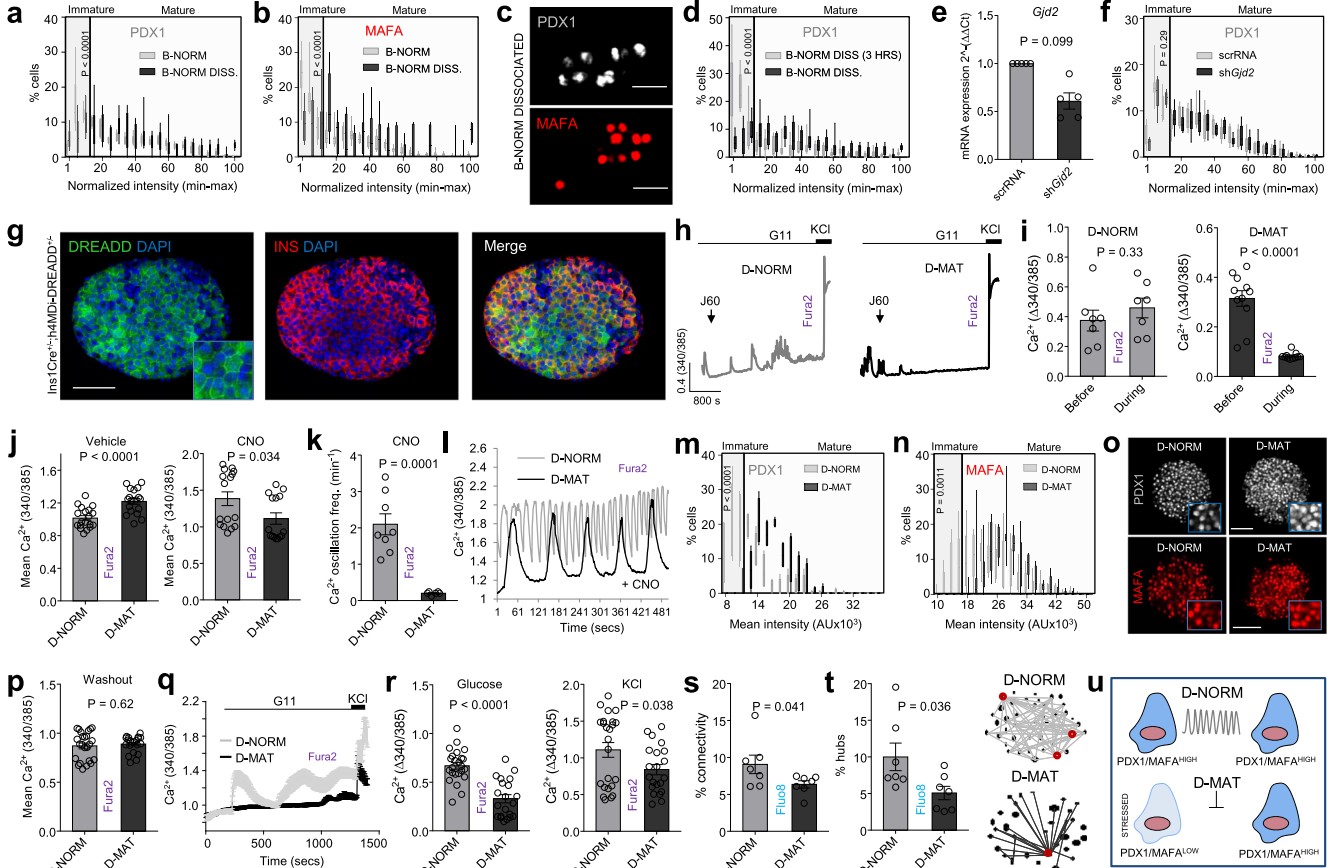

**Fig. 7 Differences in PDX1 and MAFA levels are encoded by islet signaling patterns. a–c** Islet dissociation (B-NORM DISS.) leads to loss of β-cells in the bottom 15 percentile for PDX1 (**a**) and MAFA (**b**), also shown by representative images (**c**) (B-NORM data are superimposed for comparison) (*n* = 6 islets/4 animals; two-way ANOVA, Bonferroni's multiple comparison) (PDX1: F = 7.23, DF = 19) (MAFA: F = 4.69, DF = 20) (scale bar = 42.5 μm). **d** PDX1^LOW β-cells are present 3 h following islet dissociation (*n* = 80 islets/10 coverslips; two-way ANOVA; Bonferroni's multiple comparison test) (PDX1: F = 9.54, DF = 40) (MAFA: F = 5.22, DF = 20). **e, f** sh*Gjd2* decreases *Gjd2* expression (**e**) (*n* = 5 animals; paired t-test), but this does not alter the proportion of PDX1^LOW β-cells (**f**) (*n* = 8 islets/2 animals; two-way ANOVA, Bonferroni's multiple comparison) (F = 12.85, DF = 20). **g–i** h4MDi is expressed at the β-cell membrane (**g**) (*n* = 3 islets) (scale bar = 85 μm), allowing silencing of Ca^2+ activity in D-MAT but not D-NORM (control) islets (**h, i**) (*n* = 7 islets/3 animals; paired t-test). **j** 3 h CNO incubation decreases Ca^2+ levels in D-MAT islets (vehicle, DMSO) (*n* = 16 islets/5 animals; Mann-Whitney U-test). **k, l** CNO decreases Ca^2+ oscillation frequency (**k**) in D-MAT islets, also shown by traces (**l**) (*n* = 6 islets/2 animals; unpaired t-test). **m–o** 48 h CNO incubation induces β-cell loss in the bottom 15 percentile for PDX1 (**m**) and MAFA (**n**) in D-MAT islets, also shown by representative images (**o**) (*n* = 8 islets/3 animals; two-way ANOVA, Bonferroni's multiple comparison) (PDX1: F = 5.34, DF = 20) (MAFA: F = 4.63, DF = 20) (scale bar = 60 μm). **p** 2 h washout restores Ca^2+ levels in CNO-treated islets (*n* = 21 islets/3 animals; unpaired t-test). **q, r** Ca^2+ traces (**q**) showing blunted responses to 11 mM glucose (G11) and KCl (10 mM) (**r**) in D-MAT islets (following CNO washout) (*n* = 21 islets/3 animals; unpaired t-test). **s, t** D-MAT islets display decreases in β-cell–β-cell connectivity (**s**), associated with hub loss (red circles) (**t**) (*n* = 7 islets/4 animals; Mann Whitney U-test). **u** Schematic showing effects of altering Ca^2+ signaling patterns. Bar graphs and traces show the mean ± SEM. Box-and-whiskers plot shows median and min-max. All tests are two-sided where relevant.

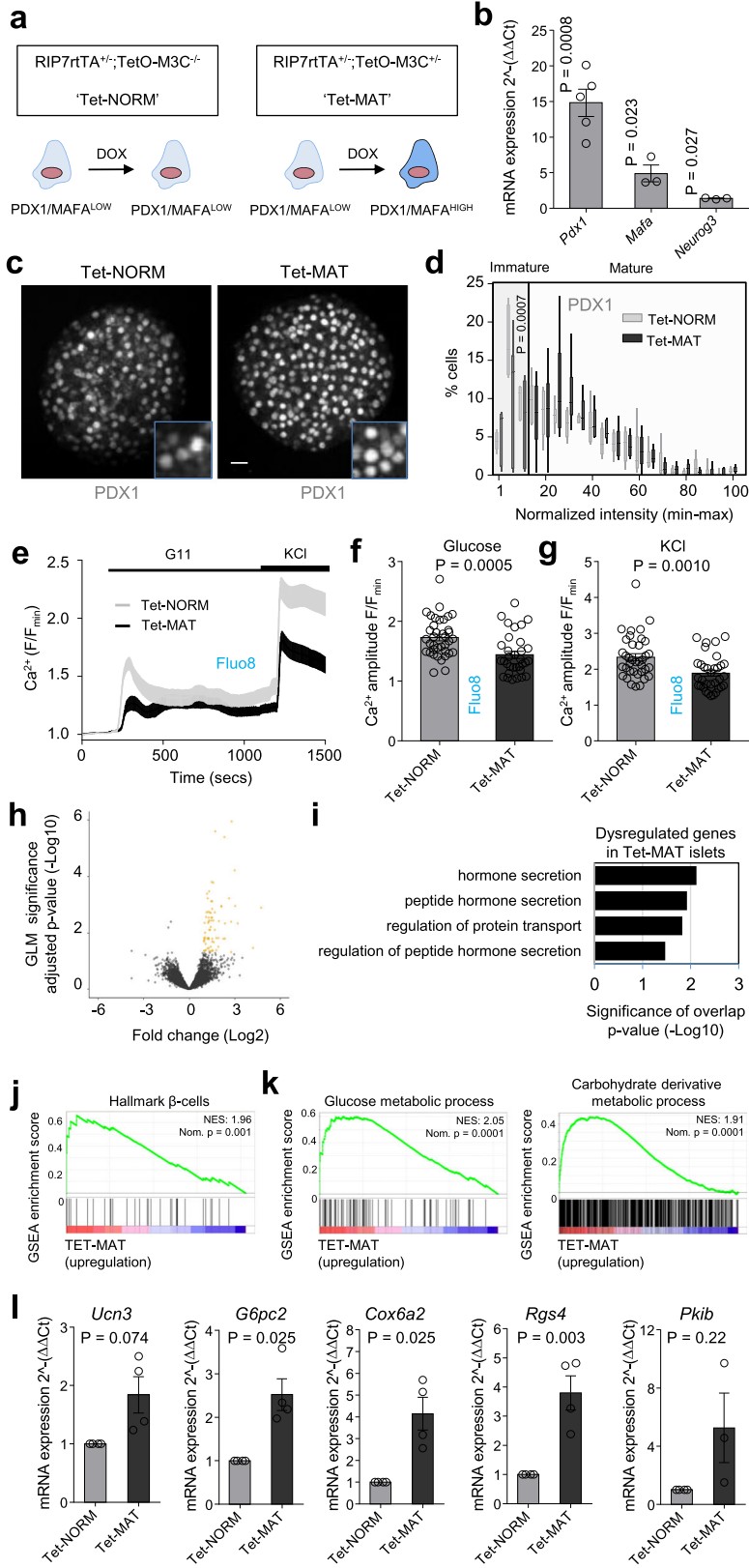

PDX1$^{HIGH}$/MAFA$^{HIGH}$ β-cells might contribute to proper islet function and insulin release.

Islets with an increased proportion of PDX1$^{HIGH}$/MAFA$^{HIGH}$ β-cells displayed a large reduction in β-cell-β-cell connectivity. This was associated with a decreased number of hubs, immature and energetic cells previously shown to coordinate glucose responsiveness[21]. Indeed, β-cells in B-MAT islets responded more stochastically to glucose, closely resembling the responses seen in islets from *ob/ob* or Cx36$^{-/-}$ animals[52–54], as well as following silencing of hubs and their associated cell clusters[21], or uncoupling

**Fig. 8 The balance of PDX1$^{LOW}$:PDX1$^{HIGH}$ β-cells influences islet gene expression. a** Recombination of RIP7rtTA and TetO/M3C mice allows doxycycline-inducible changes in β-cell *Neurog3, Pdx1* and *Mafa* expression in Tet-MAT but not Tet-NORM (control) islets. **b** *Pdx1, Mafa* and *Neurog3* expression increases following incubation of Tet-MAT islets with 100 ng/ml doxycycline for 48 h ($n = 3$ animals; paired *t*-test). **c, d** A significant decrease in the number of PDX1$^{LOW}$ β-cells is seen in doxycycline-treated Tet-MAT islets, as shown by representative images (**c**), and shown also by the loss of cells in the lowest fluorescence intensity bins (**d**) ($n = 6$ islets/3 animals; two-way ANOVA, Bonferroni's multiple comparison) (scale bar = 20 μm) (F = 41368, DF = 20). **e–g** Mean traces (**e**) and bar graphs (**f** and **g**) showing impaired glucose- and KCl-stimulated Ca$^{2+}$ rises in Tet-MAT but not Tet-NORM islets ($n = 33$ islets/4 animals; unpaired *t*-test). **h** Volcano plot of differential gene expression between Tet-NORM and Tet-MAT islets. Fold-change (Log2, *x*-axis) gene expression is plotted against adjusted *p*-value for differential gene expression (normalized by GLM, -Log10, y-axis). Colored dots represent Ensembl genes that are differentially regulated at an adjusted *p*-value < 0.05 ($n = 5$ animals). **i** Gene ontology analysis of differentially regulated genes in Tet-MAT islets. A set of 83 genes were functionally annotated using DAVID (adjusted p-value of < 0.05). **j** Gene set enrichment analysis (GSEA) suggests that genes belonging to the gene set "hallmark β-cells" are upregulated in Tet-MAT islets. Normalized enrichment score (NES) and nominal p-value is presented in the top right corner of the graph. **k** GSEA analysis shows enrichment of genes belonging to glucose and carbohydrate derivative metabolic processes amongst the upregulated genes in Tet-MAT islets. **l** RT-qPCR analyses confirming upregulation of *Ucn3, G6pc2, Cox6a2* and *Rgs4* but not *Pkib* in Tet-MAT islets ($n = 3$ animals; paired *t*-test). Bar graphs and traces show the mean ± SEM. Box-and-whiskers plot shows median and min-max. All tests are two-sided where relevant.

of β-cells following dissociation[21]. How might PDX1$^{LOW}$/MAFA$^{LOW}$ β-cells affect β-cell-β-cell coordination so profoundly? We speculate that these cells might be gap junction-coupled as a network within the islet, since mRNA for Cx36 decreased ~50% following their loss, although we acknowledge that dual patch recordings of PDX1$^{LOW}$ cells would be needed to provide definitive evidence for this. Together with the tendency of PDX1$^{LOW}$ cells to mount higher amplitude Ca$^{2+}$ rises, such preferential communication could allow a subset of β-cells to regulate excitability in neighboring β-cells, as shown by recent modeling approaches[55]. Alternatively, increases in the proportion of mature β-cells might perturb islet function by influencing gene expression or paracrine circuits such as those mediated by somatostatin and GABA. Nonetheless, these results obtained using three different models (viral transduction, DREADD and doxycycline-inducible) confirm our previous optogenetic findings on hub cells[21], and suggest that a continuum of immature β-cells exists with shared phenotypic and functional features.

While raw insulin secretion was unchanged in B-MAT versus B-NORM islets, the proportion of total insulin secreted was reduced. This secretory defect is likely due to a combination of factors reported here, including: (1) reduced glucose-stimulated metabolism (ATP/ADP); (2) decreased Ca$^{2+}$ influx, which was refractory to generic depolarizing stimulus; (3) defective β-cell-β-cell coordination; and (4) impaired glucose-induced amplifying signals (cAMP), which could not be restored with incretin mimetic or forskolin. Insulin granule density at the membrane and exocytotic marker gene expression were both unchanged.

A feature of B-MAT islets was downregulated expression of genes encoding Ca$^{2+}$ channels. Given that PDX1 and MAFA are required for β-cell Ca$^{2+}$ fluxes, what are the mechanisms involved? One possibility is that Ca$^{2+}$ channel expression is higher in PDX1$^{LOW}$/MAFA$^{LOW}$ β-cells due to a fine poise between transcription factor expression and regulation of downstream gene targets (Goldilocks effect). Indeed, recent studies have shown that patients with a stabilizing MAFA missense mutation show reduced insulin secretion[56], suggestive of defects in stimulus-secretion coupling. In addition, metabolism was altered in B-MAT islets, yet *Cox6a2*, which encodes an electron transport chain subunit, was upregulated. Unusually, however, *Cox6a2* is an ADP-binding subunit of respiratory chain complex IV, previously shown to upregulate uncoupling protein 2 expression[45]. Therefore, overexpression of *Cox6a2* would be expected to dissociate mitochondrial oxidative metabolism from ATP/ADP generation, as shown by our imaging data. Moreover, the decrease in ATP/ADP and Ca$^{2+}$ responses to glucose detected in B-MAT islets is largely consistent with previous observations showing that cells with immature traits (hubs)[21,22] are metabolically-adapted, and

that cells with low exocytosis (*RBP4*$^+$)[5] still possess normal Ca$^{2+}$ currents. Lastly, imaging of PDX$^{LOW}$ cells, triaged by expression of the BFP reporter, revealed an inverse association between PDX1 levels and Ca$^{2+}$ amplitude when viewed in the islet context.

Supporting a critical role for cell-cell interactions in driving a diverse β-cell maturity profile, experiments in dissociated islets revealed a decrease in the proportion of PDX1$^{LOW}$/MAFA$^{LOW}$ cells. The intra-islet mechanisms that support heterogeneity in β-cell maturity (and ergo the existence of PDX1$^{LOW}$/MAFA$^{LOW}$ cells) likely include Ca$^{2+}$ signaling dynamics and depolarization status, since PDX1$^{LOW}$/MAFA$^{LOW}$ cells were also reduced in chemogenetic experiments in which β-cells were conditionally perturbed. Mechanistically, Ca$^{2+}$ fluxes have been shown to suppress Ca$^{2+}$-regulated genes to impair β-cell identity[37]. Our results suggest that cells with lower levels of PDX1 and MAFA might be more sensitive to this phenomenon, since their phenotype tends to be lost when Ca$^{2+}$ dynamics are dampened in the normal islet. Following chemogenetic silencing, a decrease in the number of PDX1$^{LOW}$/MAFA$^{LOW}$ β-cells was associated with impaired Ca$^{2+}$ responses to both glucose and KCl, as for the overexpression models. By contrast, Ca$^{2+}$ responses to KCl remain intact in K$_{ATP}$ gain-of-function (GOF) islets, despite similar levels of β-cell hyperpolarization[57,58]. This difference is likely due to changes in voltage-dependent Ca$^{2+}$ channel function in D-MAT islets, which presented with decreased expression of the Ca$^{2+}$ channel subunit *Cacna1d*. It will be interesting to explore whether PDX1$^{LOW}$/MAFA$^{LOW}$ β-cells are lost in other models where depolarization status can be controlled (e.g. using K$_{ATP}$ GOF or optogenetics).

There are a number of limitations with the present study that should be noted. Firstly, while transition of PDX1/MAFA$^{LOW}$ -> PDX1/MAFA$^{HIGH}$ β-cells (and thus an increase in the proportion of mature β-cells) can be statistically inferred post-transduction, we cannot exclude a more widespread overexpression that also encompasses PDX1/MAFA$^{HIGH}$ β-cells. Secondly, impaired β-cell function in B-MAT islets might stem from loss of transcriptional dynamics. For example PDX1/MAFA$^{LOW}$ -> PDX1/MAFA$^{HIGH}$ cells might transition over the hours timescale[9,59], and clamping this using overexpression approaches might constrain insulin release. Thirdly, we cannot exclude that PDX1/MAFA$^{LOW}$ -> PDX1/MAFA$^{HIGH}$ cells become senescent or apoptotic, although neither of these possibilities are supported by our transcriptomic analyses. Also, we only looked at islets from 8–12 week-old animals and further studies are required across lifespan, as well as in response to metabolic stressors, especially since senescent β-cells possess transcriptomic signatures of immature cells[13]. Fourthly, NEUROG3 was mildly overexpressed, which could feasibly lead to a progenitor-like β-cell state. We

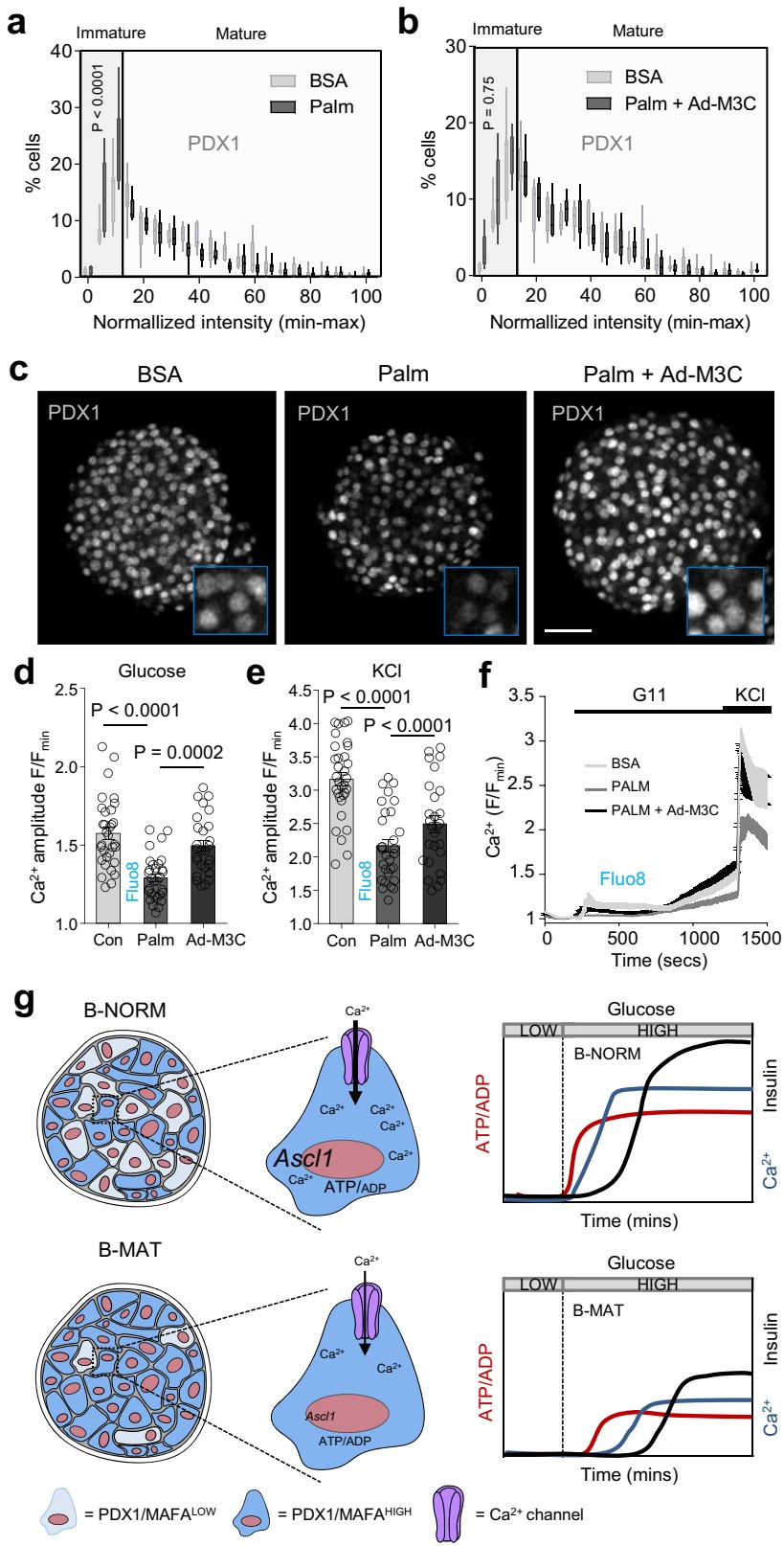

think that this is unlikely, as NEUROG3 protein was only weakly detectable, NEUROG3 exists in a dephosphorylated form in the adult islet where it helps to maintain a differentiated state[60,61], and results were replicated in a chemogenetic model that does not possess NEUROG3 activity. In addition, the transcriptomic

profile of B-MAT islets did not reveal enrichment for progenitor signatures and classically-defined β-cell identity was apparently normal.

Lastly, we acknowledge a number of potential limitations with the overexpression system, quantification and imaging approaches

**Fig. 9 Maintaining PDX1^LOW:PDX1^HIGH β-cell balance protects against islet failure. a–c** A significant decrease in the proportion of PDX1^HIGH β-cells is detected in palmitate-treated islets (**a**), and this can be reversed using Ad-M3C (**b**), as shown by representative images (**c**) ($n = 7$ islets/4 animals; two-way ANOVA, Bonferroni's multiple comparison) (Palm: F = 4.28, DF = 20) (Palm + Ad-M3C: F = 0.90, DF = 20) (BSA, bovine serum albumin; Palm, 0.5 mM palmitate for 48 h) (scale bar = 42.5 μm). Note that the same BSA-only (control) PDX1 fluorescence intensity distribution is shown in both graphs (**a**) and (**b**) to allow cross-comparison (the experiments were performed in parallel). **d–f** $Ca^{2+}$ responses to glucose (**d**) and KCl (**e**) are blunted in palmitate-treated, but not palmitate + Ad-M3C-treated islets ($n = 27$ islets/4 animals; one-way ANOVA, Sidak's multiple comparison) (G11: F = 18.80, DF = 2) (KCl: F = 23.13, DF = 2), as shown by mean traces (**f**). **g** Schematic showing that a decrease in the proportion of PDX1^LOW/MAFA^LOW β-cells leads to altered islet $Ca^{2+}$ fluxes, decreased expression of $Ca^{2+}$-dependent genes such as *Ascl1*, and broader changes to β-cell function, including impaired ATP/ADP and insulin responses to glucose. Bar graphs and traces show the mean ± SEM. Box-and-whiskers plot shows median and min-max. All tests are two-sided where relevant.

used here: 1) generalized transcription factor overexpression, especially that involving NEUROG3, might lead to impaired islet function and insulin secretion; 2) underestimation of overexpression skewed toward the highest PDX1/MAFA signal intensity bins cannot be excluded (i.e. it might be more difficult to detect overexpression in a cell that already has high levels); 3) imaging could suffer from technical noise, decreasing our ability to accurately quantify PDX1 and MAFA; and 4) exogenous PDX1 might possess different activity or affect different targets compared to endogenous PDX1. In addition, we cannot exclude that inter-cellular feedback is present whereby when more cells express PDX1 and MAFA, those expressing the highest levels make less, as suggested by the RT-qPCR analyses of endogenous *Pdx1* levels. Further studies using surface markers and lineage labels, together with scRNA-seq or spatial transcriptomics, will be needed to categorically confirm overexpression specifically in immature β-cells in the intact tissue.

In summary, we have performed an in-depth functional interrogation of islets in which proportionally more β-cells have been made mature in terms of PDX1 and MAFA expression levels. These studies suggest that proper islet function is dependent on the co-existence of PDX1^LOW/MAFA^LOW and PDX1^HIGH/MAFA^HIGH β-cells in the tissue context. Findings from single-cell screening studies or studies in dissociated cells should thus be interpreted carefully in light of differences imparted by the tissue context.

## Methods

**Mouse models**. Wild type CD1, Ins1Cre^Thor knock-in (IMSR Cat# JAX:026801, RRID:IMSR_JAX:026801), Ins1CreERT knock-in (IMSR Cat# JAX:026802, RRID: IMSR_JAX:026802)[62] or Pdx1-BFP fusion mice[26] were used as tissue donors for overexpression experiments with adenovirus containing a polycistronic construct for CMV-NEUROG3/PDX1/MAFA/mCherry (Ad-M3C)[23,24]. Pdx1-BFP animals contain blue fluorescent protein (BFP) fused to the open reading frame (ORF) of PDX1 in which the STOP codon in exon 2 has been deleted. Pdx1-BFP mice are viable and fertile without signs of MODY4, and BFP fluorescence reflects endogenous PDX1 levels[26].

To allow identification of non-β-cells, Ins1Cre^Thor animals were crossed with the R26^mTmG reporter strain (IMSR Cat# JAX:007576, RRID:IMSR_JAX:007576), resulting in Ins1Cre^Thor+/−; R26^mTmG−fl/− animals harboring Cre-dependent excision of tdTomato.

Chemogenetic constructs were conditionally expressed in β-cells by crossing Ins1Cre animals[62] with those possessing flox'd alleles for the mutant muscarinic receptor hM4Di (IMSR Cat# JAX:026219, RRID:IMSR_JAX:026219)[63]. The presence of Cre was accounted for by using Ins1Cre^Thor+/−;hM4Di-DREADD^fl/− (D-MAT) and Ins1Cre^Thor+/−;hM4Di-DREADD^−/− (D-NORM) littermates.

To achieve conditional overexpression of M3C, mice harboring the tetracycline trans-activator under the control of the Ins2 promoter (RIP7rtTA)[64] were crossed with animals engineered to possess M3C upstream of a tetracycline response element (M3C-TetON)[25]. Littermate controls contained the RIP7rtTA allele, given previously reported issue with *Ins2* constructs (RIP7rtTA^+/−;TetO-M3C^+/− and RIP7rtTA^+/−;TetO-M3C^−/−, termed Tet-MAT and Tet-NORM, respectively)[64].

Male and female 6–12 week-old animals were maintained in a specific pathogen free facility, with free access to food. Environmental conditions were: 21 ± 2 ˚C, 55 ± 10% relative humidity and a 12 hr light-dark cycle. Animal studies were regulated by the Animals (Scientific Procedures) Act 1986 of the United Kingdom and performed under Personal Project Licence P2ABC3A83. Approval was granted by the University of Birmingham's Animal Welfare and Ethical Review Body and all ethical guidelines were adhered to whilst carrying out this study.

**Human donors**. Human islets were obtained from brain-dead or deceased human donors from isolation centers in Canada (Alberta Diabetes Institute, IsletCore) and Italy (San Raffaele, Milan). Procurement of human islets was approved by the Human Research Ethics Board (Pro00013094, Pro00001754) at the University of Alberta, as well as the Ethics Committee of San Raffaele Hospital in Milan, with written informed consent from next of kin. Studies with human islets were approved by the University of Birmingham Ethics Committee, as well as the National Research Ethics Committee (REC reference 16/NE/0107, Newcastle and North Tyneside, U.K.). Anonymized Donor characteristics are reported in Supplementary Table 1.

**Islet isolation**. Mice were euthanized by cervical dislocation before inflation of the pancreas via injection of collagenase solution (1 mg/ml; Serva NB8) into the bile duct. Pancreata were then digested for 12 mins at 37 °C in a water bath before purification of islets using a Histopaque or Ficoll gradient. Islets were hand-picked and cultured (5% $CO_2$, 37 °C) in RPMI medium containing 10% FCS, 100 units/mL penicillin, and 100 μg/mL streptomycin.

**Human islet culture**. Human islets were cultured (5% $CO_2$, 37 °C) in: CMRL supplemented with 10% FCS, 100 units/mL penicillin, 100 μg/mL streptomycin, 0.25 μg/mL fungizone and 5.5 mmol/L D-glucose, or low glucose DMEM supplemented with 10% FCS, 100 units/mL penicillin and 100 μg/mL streptomycin. For donor details, see Table S1.

**Loss of PDX1^LOW/MAFA^LOW and PDX1^HIGH/MAFA^HIGH β-cells**. WT islets were transduced for 48–72 h with Ad-M3C construct[24]. Ad-PATagRFP[21] was used to confirm absence of off-target effects. For knockdown of PDX1, mCherry-tagged short hairpin RNA against *Pdx1* (sh*Pdx1*) or scrambled control (scrRNA) were delivered using adenovirus (Vector Biolabs Cat# shADV-268353 and 1122). Expression levels were verified with RT-qPCR using SYBR Green or TaqMan chemistry with primers and probes against viral and native *Neurog3/NEUROG3*, *Pdx1/PDX1* and *Mafa/MAFA*[24]. Additionally, experiments were repeated using islets isolated from Pdx1-BFP fusion mice[26]. Islets from RIP7rtTa^+/−;M3C-TetON^+/− and RIP7rtTa^+/−;M3C-TetON^−/− mice were incubated with doxycycline 100 ng/ml for 48 h to induce transgene expression.

**Gene expression-mRNA levels**. Quantitative real-time PCR (RT-qPCR) was performed on Applied Biosystems 7500 and QuantStudio 5 instruments using PowerUp SYBR Green Master Mix (Thermo Fisher Scientific Cat# A25742), or Taqman Fast Advanced Master Mix (Thermo Fisher Scientific Cat# 4444557). Fold-change in mRNA expression was calculated compared with *Actb/Gapdh/Ppia* by using the $2^{-\Delta\Delta Ct}$ method. Gap junction knock down was achieved using adenoviral particles harboring either *Gjd2* shRNA or scrambled control (Vector Biolabs)[21]. For primer and probe details, see Supplementary Table 2.

**Immunostaining**. Islets were incubated overnight at 4 °C with primary antibodies against MAFA (Bethyl Cat# IHC-00352, RRID:AB_1279486, dilution 1:200), PDX1 (mouse: DSHB Cat# F6A11, RRID:AB_1157904, dilution 1:500 and human: Abcam Cat# ab47308, RRID:AB_777178, dilution 1:10000), NEUROG3 (DSHB Cat# F25A1B3, RRID:AB_528401, dilution 1:50) insulin (Cell Signaling Technology Cat# 3014, RRID:AB_2126503, dilution 1:500), glucagon (Sigma-Aldrich Cat# G2654, RRID:AB_259852, dilution 1:2000), somatostatin (Thermo Fisher Scientific Cat# 14-9751-80, RRID:AB_2572981, dilution 1:5000), glucokinase (Santa Cruz Biotechnology Cat# sc-7908, RRID:AB_2107620, dilution 1:50), GLP1R (DSHB Cat# Mab 7F38, RRID:AB_2618101, dilution 1:15), GFP (Aves Lab Cat# 1020, RRID:AB_10000240, dilution 1:500) and PCNA (Cell Signaling Technology Cat# 2586, RRID:AB_2160343, dilution 1:2400), before washing and application of either Alexa/DyLight 488 (Thermo Fisher Scientific Cat# A-11029, RRID: AB_2534088 and Thermo Fisher Scientific Cat# SA5-10038, RRID:AB_2556618),

Alexa 568 (Thermo Fisher Scientific Cat# A10042, RRID:AB_2534017) and Alexa/DyLight 633 (Thermo Fisher Scientific Cat# A-21052, RRID:AB_2535719 and Thermo Fisher Scientific Cat# 35513, RRID:AB_1965952) secondary antibodies. All secondary antibodies were diluted 1:1000. To avoid cross-reactivity between antibodies from the same species, sequential staining and re-blocking was performed. Samples were mounted on coverslips containing VECTASHIELD HardSet with DAPI (Vector Laboratories Cat# H-1500).

Imaging was performed using Zeiss LSM780/LSM880 confocal microscopes equipped with 25x / 0.8 / water, 40x / 1.2 / water, 100x / 1.46 / oil objectives and operated using Zen Black Edition (Zeiss, version 8.1.0.484). Super-resolution imaging was performed using the Airyscan module of the LSM880 (~140 nm). Excitation was delivered using λ = 405 nm, 488 nm, 561 nm and 633 nm laser lines. Signals were detected at λ = 428–481 nm (DAPI), λ = 498–551 nm (Alexa/DyLight 488), λ = 577–621 nm (Alexa568) and λ = 641–739 nm (Alexa/Dylight 633) using highly-sensitive GaAsP spectral detectors. A subset of experiments was performed using a Leica TCS SP5 confocal equipped with a 63x / 1.3 / glycerol objective and HyD detectors (operated with Leica Application Suite X, version 2.7). Quantification of PDX1 and MAFA staining was performed using a custom routine in ImageJ. Briefly, Gaussian filtered images were subjected to an auto-threshold and binarization step to create a mask, which was then used to identify mean pixel intensity in each PDX1+ or MAFA+ cell before construction of a frequency distribution. Glucokinase, insulin, glucagon and somatostatin were quantified using corrected total cell fluorescence (CTCF), according to the following equation: CTCF = integrated density – (area of ROI x mean fluorescence of background). Images were de-noised using a Gaussian smoothing procedure, and linear adjustments to brightness and contrast were made for presentation purposes.

**Live imaging**. For $Ca^{2+}$ imaging, islets were loaded with Fluo8 (AAT Bioquest Cat# 21082-AAT) or Fura2 (HelloBio HB0780-1mg), or transduced with Ad-GCaMP6$^m$, before imaging using a Crest X-Light spinning disk system coupled to a Nikon Ti-E base and 10 x / 0.4 / air or 25 x / 0.8 / air objective (operated using Molecular Devices Metamorph version 7.10.3). In Fluo8 experiments, excitation was delivered at λ = 458–482 nm using a Lumencor Spectra X light engine, with emitted signals detected at λ = 500–550 nm using a Photometrics Delta Evolve EM-CCD. For experiments with the ratiometric $Ca^{2+}$ indicator, Fura2, excitation was delivered at λ = 340 nm and λ = 385 nm using a FuraLED system, with emitted signals detected at λ = 470–550 nm.

ATP/ADP imaging was performed as for Fluo8, except islets were infected with adenovirus harboring the ATP/ADP sensor, Perceval[65] (a kind gift from Prof. Gary Yellen, Harvard), for 48 h. For cAMP imaging, islets were infected with adenovirus harboring Epac2-camps (a kind gift from Prof. Dermot Cooper, Cambridge). Excitation was delivered at λ = 430–450 nm and emission detected at λ = 460–500 and λ = 520–550 nm for Cerulean and Citrine, respectively. Fura2 and Epac2-camps intensity were calculated as the ratio of 340/385 or Cerulean/Citrine, respectively. Traces were presented as raw or F/F$_{min}$ where F = fluorescence at any timepoint and F$_{min}$ = minimum fluorescence.

Ins1Cre;R26$^{mTmG}$ islets were transduced with Ad-M3C before live imaging using a Zeiss LSM780 meta-confocal microscope, as above. mGFP, tdTomato and mCherry were excited with λ = 488 nm, 561 nm and 594 nm laser lines. Excitation was collected at λ = 498–551 nm (mGFP), λ = 573–590 nm (tdTomato) and λ = 603–691 nm (mCherry).

HEPES-bicarbonate buffer was used, containing (in mmol/L) 120 NaCl, 4.8 KCl, 24 NaHCO₃, 0.5 Na₂HPO₄, 5 HEPES, 2.5 CaCl₂, 1.2 MgCl₂, and 3–17 D-glucose. All microscope images were analyzed using ImageJ (NIH, version 1.5j8), Zen 2012 Blue Lite (Zeiss, version 1.1.2.0) and Zen Black Edition (Zeiss, version 8.1.0.484).

**Western blotting**. Samples were collected in urea Laemmli sample buffer (0.2 M Tris-HCl, pH 6.8, 40% glycerol, 8% SDS, 5% B-ME, 6 M Urea, 0.005% Bromophenol Blue) and sonicated (2 × 5 s at 20 kHz). Proteins were separated by SDS–PAGE (10% Acrylamide Bis-Tris Gel) with MOPS-SDS running buffer and transferred on to PVDF membranes. Membranes were blocked with TBS-T buffer (Tris-Buffered Saline containing 0.1% Tween-20) containing 5% (w/v) non-fat skimmed milk powder for 1 h at room temperature. Membranes were then incubated in antibodies against PDX1 (Iowa DSHB Cat#F6A11, RRID:AB_1157904, dilution 1:1000) and GAPDH (Cell Signaling Technology Cat# 5174, RRID: AB_10622025, dilution 1:2000), diluted in TBS-T containing 3% (w/v) bovine serum albumin (BSA) overnight at 4 °C. Membranes were washed 3 × 10 mins in TBS-T followed by incubation in horseradish peroxidase-conjugated (HRP-conjugated) secondary antibodies (anti-rabbit IgG, Cell Signaling Technology Cat# 7074, RRID:AB_2099233, dilution 1:5000 and anti-mouse IgG, Cell Signaling Technology Cat# 7076, RRID:AB_330924, dilution 1:5000) in TBS-T for 1 h at room temperature. Membranes were washed for a further 3 × 10 mins in TBS-T. ECL western blotting detection reagent (Millipore Cat# WBKLS0500) was used as per manufacturer's instructions to expose images followed by capture on G-Box (SynGene Chemi XR5). Due to signal overlap with most of the commonly used loading controls, samples were run in parallel on separate gels before immunoblotting for PDX1 and GAPDH. Full blot scans for all experiments are shown in Supplementary Figure 8.

**Insulin secretion measures**. Mouse: batches of 10 mouse islets were acclimatized in low protein-bind 1.5 ml Eppendorf tubes containing 0.5 ml HEPES-bicarbonate buffer supplemented with 3 mM glucose and 0.1% BSA. Buffer was then removed before addition of either 3 mM glucose, 16.7 mM glucose or 16.7 mM glucose + 20 nM Exendin-4 (AnaSpec Cat# ANA24463), and incubation for 30 min at 37 °C. Human: batches of 15 human islets were stimulated with 3 mM, glucose, 16.7 mM glucose or 16.7 mM glucose + 20 nM Exendin-4 according to IsletCore protocols IO (Static Glucose-stimulated Insulin Secretion (GSIS) Protocol - Human Islets V.2). Total insulin was extracted using acid ethanol and insulin concentration determined using an ultra-sensitive HTRF assay (Cisbio Cat# 62IN2PEG) according to the manufacturer's instructions. In all cases, values are normalized against total insulin for each individual experiment to account for differences in β-cell proportion with treatment and islet size.

**Chemogenetics**. The h4MDi ligands JHU37160 (J60) (Hello Bio Cat# HB6261) and clozapine N-oxide (CNO) (Tocris Cat# 4936/10) were applied to islets at 1 μM for the indicated time points. While P450 converts CNO into clozapine, which promiscuously binds endogenous receptors in vivo[66], this is not expected to be an issue in vitro. In any case, CNO was present under all conditions examined to account for off-target effects. For assessment of intraislet insulin signaling, control islets were treated with 50 nM insulin receptor antagonist S961 (Phoenix Pharmaceuticals, Cat# 051-56) for 48 h.

**Next generation sequencing**. Sequencing libraries were prepared using RNA (RIN > 7) with the Lexogen Quantseq3 FWD kit (Lexogen Cat# 015.24). Libraries were sequenced using HiSeq2000 across a single flowcell generating 75 bp long single ended reads (Illumina Cat# 20024904). All samples were prepared and sequenced as a single pool. Trimmomatic software (v0.32) and bbduk.sh script (Bbmap suite) was used to trim the ILLUMINA adapters, polyA tails and low-quality bases from reads. Trimmed reads were then uniquely aligned to the human genome (hg38) using STAR (v2.5.2b) and the Gencode (v28, Ensembl release 92) annotation as the reference for splice junctions. Between 4–6 M mapped reads per sample were quantified using HT-seq (v0.9.1) using Gencode (v28) genes (-intersection-nonempty flag).

**Correlation and wavelet analyses**. Detection of superconnected islet regions was performed using matrix binarization analyses developed in-house[67]. Cells were identified using a region of interest (ROI), intensity over time traces extracted, subjected to Hilbert-Huang empirical mode decomposition to remove noise and baseline trends, and a 20% threshold imposed to binarize cells according to activity status. Co-activity between all cell pair combinations was assessed using the equation $C_{ij} = \frac{T_{ij}}{\sqrt{T_i T_j}}$ where C is a correlation coefficient, $T_i$ and $T_j$ is the period spent ON for each cell, and $T_{ij}$ is the period both cells spend ON together. Significance was calculated versus the randomized dataset for each cell pair using a permutation step for each binarized data row. This analysis allows identification of cells whose activity repetitively spans that of the rest of the population. Superconnected cells or hubs were defined as cells possessing 60–100% of the correlated links and plotted on functional connectivity maps using the Euclidean coordinates.

Wavelet analysis was used to determine the time-localized $Ca^{2+}$ oscillation frequency. Spectra were extracted from $Ca^{2+}$ traces with a univariate bias-corrected wavelet transform (biwavelet package in R), which prevents compression of power as period lengthens. Period was then depicted against time, with a color ramp representing frequency power.

**Differential gene expression analyses**. Differential gene expression was obtained using DEseq2 with age- and sex-matched paired Tet-NORM (n = 5) and Tet-MAT samples (n = 5). Differentially expressed genes between control and Tet-MAT islets at adjusted p-value <0.05 were annotated using DAVID BP_FAT, with high stringency for clustering.

Gene set enrichment analysis (GSEA) was used to interrogate specific gene sets against expression data. GSEA calculates an Enrichment Score (ES) by scanning a ranked-ordered list of genes (according to significance of differential expression (-log10 p-value)), increasing a running-sum statistic when a gene is in the gene set and decreasing it when it is not. The top of this list (red) contains genes upregulated in Tet-MAT islets while the bottom of the list (blue) represents downregulated genes. Each time a gene from the interrogated gene set is found along the list, a vertical black bar is plotted (hit). If the hits accumulate at the bottom of the list, then this gene set is enriched in downregulated genes (and vice versa). If interrogated genes are distributed homogenously across the rank-ordered list of genes, then that gene set is not enriched in any of the gene expression profiles. We converted human gene sets into homologous mouse gene sets using the homologous gene database from MGI.

Analysis of published human single cell data[31,33] was performed using Monocle (http://cole-trapnell-lab.github.io/monocle-release/).

**Statistics and reproducibility**. All analyses were conducted using GraphPad Prism (version 8.4.3), Igor Pro (Wavemetrics, version 5.05 A), R Studio (R Project, version 1.1.456) or MATLAB (Mathworks, version R2018b) software. Unpaired or

paired Student's *t*-test was used for pairwise comparisons. Multiple interactions were determined using normal or repeated measures ANOVA followed by Bonferroni or Sidak posthoc testing (accounting for degrees of freedom). Straight lines were fitted with linear regression whilst a polynomial trend was used for multiple regression. Goodness of fit was calculated using $R^2$. Details of replicate nature and number can be found in the figure legends. Where the n number occupies a range of samples (e.g., 20–24 islets) or animals (e.g., 3–5 animals), the lowest value is provided as per journal guidelines. Hence, the number of datapoints on the graph may be higher than the stated sample size. When representative images are provided, the experiment was repeated the same number of times as the related quantification, with similar results.

**Reporting summary**. Further information on research design is available in the Nature Research Reporting Summary linked to this article.

## Code availability

Bioinformatic pipelines used to analyze RNA-sequencing data can be found at https://github.com/iakerman/QuantSeq (https://doi.org/10.5281/zenodo.4091681)[68].

## Data availability

Raw image files are available upon reasonable request. Raw read files and processed data files for RNA-sequencing experiments from this study can be found at the NCBI Gene Expression Omnibus (GEO) database (GSE133798). Publicly available transcriptomic data were obtained from isletregulome.org[31,32], single cell data was downloaded from https://www.ebi.ac.uk/arrayexpress/experiments/E-MTAB-5061/ and https://www.ebi.ac.uk/arrayexpress/experiments/E-MTAB-5060/ [33]. Source data are provided with this paper.

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

## Acknowledgements

We thank Dr. Jocelyn E. Manning Fox and Prof. Patrick E. MacDonald for provision of human islets via the Alberta Diabetes Institute IsletCore at the University of Alberta in Edmonton with the assistance of the Human Organ Procurement and Exchange (HOPE) program, Trillium Gift of Life Network (TGLN) and other Canadian organ procurement organizations. We are grateful to the European Consortium for Islet Transplantation (ECIT), which was supported by JDRF award 31-2008-416 (ECIT Islet for Basic Research program). D.J.H. was supported by a Diabetes UK R.D. Lawrence (12/0004431) Fellowship, a Wellcome Trust Institutional Support Award, and MRC (MR/N00275X/1 and MR/S025618/1) and Diabetes UK (17/0005681) Project Grants. This project has received funding from the European Research Council (ERC) under the European Union's Horizon 2020 research and innovation program (Starting Grant 715884 to D.J.H.). G.A.R. was supported by Wellcome Trust Senior Investigator (WT098424AIA) and Investigator (212625/Z/18/Z) Awards, MRC Programme Grants (MR/R022259/1, MR/J0003042/1, MR/L020149/1) and Experimental Challenge Grant (DIVA, MR/L02036X/1), MRC (MR/N00275X/1), Diabetes UK (BDA/11/0004210, BDA/15/0005275, BDA 16/0005485) and Imperial Confidence in Concept (ICiC) Grants. This project has received funding from the European Union's Horizon 2020 research and innovation programme via the Innovative Medicines Initiative 2 Joint Undertaking under grant agreement No 115881 (RHAPSODY) to G.A.R. L.P. provided human islets through collaboration with the Diabetes Research Institute, IRCCS San Raffaele Scientific Institute (Milan), within the European Consortium for Islet Transplantation islet distribution program for basic research supported by JDRF (1-RSC-2014-90-I-X). We thank the Microscopy and Imaging Services (MISBU) in the Tech Hub facility at Birmingham University for support and maintenance of microscopes.

## Author contributions

D.N. and D.J.H. devised the studies. D.N., N.H.F.F., F.B.A., F.C., K.V., P.D., Y-C.L, M.B., A.B-P, R.F., I.A. and D.J.H. performed experiments and analyzed data. G.S., A.D. and I.A. provided bioinformatics and analyzed data. G.A.R. and H.L. provided mice, constructs and discussed data. R.N. and L.P. isolated and provided human islets. Q.Z. provided constructs and mice. D.J.H. supervised the work. D.N. and D.J.H. wrote the paper with input from all the authors.

## Competing interests

G.A.R. has received grant funding from Servier and is a consultant for Sun Pharma. The remaining authors declare no competing interests.
