## [Peer Review File · Nature Communications]

Editorial Note: This manuscript has been previously reviewed at another journal that is not operating a transparent peer review scheme. This document only contains reviewer comments and rebuttal letters for versions considered at *Nature Communications* .

Reviewer #2 (Remarks to the Author):

The authors have revised the manuscript in such a way that the readers will be informed of the possible caveats of the study and at the same time enjoy a very thorough investigation. This manuscript should definitely be published.

Reviewer #4 (Remarks to the Author):

This resubmitted manuscript adds several cautious statements to the results and discussion sections. They have clearly shown that transcription factor overexpression and knockdown alters the proportion of PDX1^{low} and PDX1^{high} cells, and this impairs insulin secretion and cellular population dynamics.

The experiments are nicely executed but, in spite of the cautious wording in the main text, the conclusions in the title and abstract are unchanged. The following examples taken from the abstract illustrate how the authors still conclude that the number of mature cells has a causal role:

“Here we show that differences in β -cell maturity, defined using PDX1 and MAFA expression, are required for proper islet operation”

“At the transcriptomic level, the presence of increased numbers of mature β -cells led to dysregulation of gene pathways involved in metabolic processes”.

“During metabolic stress, islet function could be restored by redressing the balance between immature and mature β -cells”

Title “ Mature and immature β -cells both contribute to islet function and insulin release” Authors have not selectively eliminated mature cells or immature cells.

Ideally to make these statements one would like to eliminate mature cells or create more mature cells without affecting other functions. This study reports experiments such as overexpressing NEUROG3 PDX1 + MAFA with a tetracycline controlled promoter, which could lead to increased maturation as well as other effects. Likewise, RNA knockdown for PDX1 can have various cellular effects besides increasing the number of immature cells. This has been discussed in previous revisions, and it is clear that the authors feel strongly that this is how they wish to interpret their data, although it would seem more appropriate to describe the evidence and only then suggest an explanation.

Minor points:

The following statement in the abstract is vague, what is “the islet signalling network”?

“the islet signalling network was found to contribute to differences in maturity across β -cells”

Check abbreviations, e.g. define B-IMMAT islets. scRNA means single cell RNA-seq in the discussion, or scrambled control guide RNAs in the main text

RESPONSE TO REVIEWERS

We are delighted that our manuscript is acceptable in principle for publication in *Nature Communications*. We would like to thank the expert Reviewers and Editor for their constructive feedback during the transfer process. As requested, we have responded to the remaining minor issues raised by Reviewer 4, as well as formatted the manuscript according to the editorial requests. A point-by-point rebuttal is provided below.

REVIEWER 4

This resubmitted manuscript adds several cautious statements to the results and discussion sections. They have clearly shown that transcription factor overexpression and knockdown alters the proportion of PDX1^{low} and PDX1^{high} cells, and this impairs insulin secretion and cellular population dynamics.

The experiments are nicely executed but, in spite of the cautious wording in the main text, the conclusions in the title and abstract are unchanged. The following examples taken from the abstract illustrate how the authors still conclude that the number of mature cells has a causal role:

“Here we show that differences in β -cell maturity, defined using PDX1 and MAFA expression, are required for proper islet operation”

“At the transcriptomic level, the presence of increased numbers of mature β -cells led to dysregulation of gene pathways involved in metabolic processes”.

“During metabolic stress, islet function could be restored by redressing the balance between immature and mature β -cells”

*Title “ Mature and immature β -cells both contribute to islet function and insulin release”
Authors have not selectively eliminated mature cells or immature cells.*

Ideally to make these statements one would like to eliminate mature cells or create more mature cells without affecting other functions. This study reports experiments such as overexpressing NEUROG3 PDX1 + MAFA with a tetracycline controlled promoter, which could lead to increased maturation as well as other effects. Likewise, RNA knockdown for PDX1 can have various cellular effects besides increasing the number of immature cells. This has been discussed in previous revisions, and it is clear that the authors feel strongly that this is how they wish to interpret their data, although it would seem more appropriate to describe the evidence and only then suggest an explanation.

The Reviewer's point is well taken and we have now modified the title and abstract accordingly. In addition, we have harmonized the title, abstract and main text such that we clearly refer to PDX1^{LOW}/MAFA^{LOW} and PDX^{HIGH}/MAFA^{HIGH} cells rather than 'immature' and 'mature'. As such, we are confident that the experimental data are interpreted in line with the evidence (i.e. we clearly showed changes in proportions of PDX1^{LOW}/MAFA^{LOW} and PDX^{HIGH}/MAFA^{HIGH} cells, as the reviewer mentions).

Minor points:

The following statement in the abstract is vague, what is “the islet signalling network”?

“the islet signalling network was found to contribute to differences in maturity across β -cells”

We have now clarified this sentence as follows: “Using a chemogenetic disruption strategy, differences in PDX1 and MAFA expression are shown to depend on islet Ca^{2+} signalling patterns”.

Check abbreviations, e.g. define B-IMMAT islets. scRNA means single cell RNA-seq in the discussion, or scrambled control guide RNAs in the main text

Thanks for pointing out these issues, now corrected as follows: “Islets with proportionally more PDX1^{LOW} β -cells (B-IMMAT)”, “scrambled control (scrRNA)”.